# BRAID: Input-driven nonlinear dynamical modeling of neural-behavioral data

**Parsa Vahidi[1], Omid G. Sani[1], Maryam M. Shanechi[1-3],***
[1]Electrical and Computer Engineering, [2]Computer Science, [3]Biomedical Engineering
Viterbi School of Engineering, University of Southern California (USC), Los Angeles, CA
{pvahidi, omid.ghasemsani, shanechi}@usc.edu, *Corresponding author

## Abstract

Neural populations exhibit complex recurrent structures that drive behavior, while continuously receiving and integrating external inputs from sensory stimuli, upstream regions, and neurostimulation. However, neural populations are often modeled as autonomous dynamical systems, with little consideration given to the influence of external inputs that shape the population activity and behavioral outcomes. Here, we introduce BRAID, a deep learning framework that models nonlinear neural dynamics underlying behavior while explicitly incorporating any measured external inputs. Our method disentangles intrinsic recurrent neural population dynamics from the effects of inputs by including a forecasting objective within input-driven recurrent neural networks. BRAID further prioritizes the learning of intrinsic dynamics that are related to a behavior of interest by using a multi-stage optimization scheme. We validate BRAID with nonlinear simulations, showing that it can accurately learn the intrinsic dynamics shared between neural and behavioral modalities. We then apply BRAID to motor cortical activity recorded during a motor task and demonstrate that our method more accurately fits the neural-behavioral data by incorporating measured sensory stimuli into the model and improves the forecasting of neural-behavioral data compared with various baseline methods, whether input-driven or not.

## 1 Introduction

Understanding the relationship between neural activity and behavior is a critical goal in neuroscience and neurotechnology. Neural activity and its temporal structure, or *"dynamics"*, during a behavior are formed by the interplay between (1) the recurrent networks within a brain area, i.e., *intrinsic dynamics*, and the (2) temporally-structured inputs it receives during the behavior (Remington et al., 2018; Vyas et al., 2020). A neural population may receive inputs from measurable sources such as sensory stimuli, electrical/optogenetic neurostimulation (Buonomano & Maass, 2009; Seely et al., 2016; Susilaradeya et al., 2019; Sauerbrei et al., 2020; Shenoy & Kao, 2021; Yang et al., 2021b; Vahidi et al., 2024), as well as from other upstream brain areas (Sauerbrei et al., 2020; Shenoy & Kao, 2021), which could be included in multi-regional recordings (Jun et al., 2017; Steinmetz et al., 2019). However, even easily measurable external inputs (e.g., sensory stimuli) are often not explicitly considered when modeling neural-behavioral activity, which can lead to a conflation of intrinsic and input-driven contributions, creating challenges for interpretation (Seely et al., 2016; Sauerbrei et al., 2020; Vahidi et al., 2024). Beyond disentangling intrinsic dynamics from input dynamics, incorporating measured inputs into models can also enhance the behavior decoding performance in neurotechnologies such as stimulation-based closed-loop controllers (Yang et al., 2018; 2021b).

Another challenge is to disentangle neural dynamics that are relevant to a specific behavior from other neural dynamics, and to prioritize the former. This is critical as the majority of neural variance may not be relevant to the behavior of interest (Churchland et al., 2012; Mante et al., 2013; Kobak et al., 2016; Allen et al., 2019; Engel & Steinmetz, 2019; Stringer et al., 2019; Sani et al., 2021). While most prior works use unsupervised methods when modeling neural activity as latent variable dynamical systems (Aghagolzadeh & Truccolo, 2015; Gao et al., 2016; Wu et al., 2017; Pandarinath et al., 2018; Hernandez et al., 2020; Rutten et al., 2020; Kim et al., 2021), recent works have shown improved learning of behaviorally relevant neural dynamics by using behavior data during learning

in a supervised manner (Sani et al., 2021; Hurwitz et al., 2021; Kramer et al., 2022; Gondur et al., 2024; Vahidi et al., 2024; Abbaspourazad et al., 2024; Sani et al., 2024; Oganesian et al., 2024).

Yet another challenge is posed by the nonlinearities in neural-behavioral data. While linear models have been extremely effective in approximating neural dynamics (Hastie et al., 2009; Churchland et al., 2012; Mante et al., 2013; Cunningham & Yu, 2014; Kao et al., 2015; Kobak et al., 2016; Abbaspourazad et al., 2021; Sani et al., 2021), they may require higher dimensional latent representations compared to nonlinear models (Yang et al., 2019; Nozari et al., 2024; Sani et al., 2024), and do not provide interpretability for nonlinear dynamical phenomena such as multi-stable fixed points and limit cycles (Kim et al., 2021; Durstewitz et al., 2023). Moreover, unlike linear models, for nonlinear models the relationship between the intrinsic dynamics and an inference model that is fitted to estimate the latent states from observations is not analytically known, posing a challenge for studying intrinsic dynamics (see section 3.1).

Here, we address all aforementioned challenges by introducing Behaviorally Relevant Analysis of Intrinsic Dynamics (BRAID), a new method with the following key contributions. *First*, BRAID captures complex nonlinear structures in neural-behavioral-input data, offering greater expressivity than linear methods. *Second*, by optimizing multi-step-ahead forecasts of neural-behavior data, BRAID simultaneously learns two representations for neural dynamics: the predictor and the generative form representations (see section 3.1), the latter of which describes intrinsic dynamics. *Third*, by explicitly modeling the influence of measured inputs, BRAID disentangles their dynamics from intrinsic dynamics to more closely reflect the neuronal networks within the recorded brain region. *Fourth*, we introduce a multi-stage learning framework that dissociates and prioritizes the learning of intrinsic behaviorally relevant neural dynamics, while considering measured inputs (see section 3.2). *Fifth*, we introduce additional preprocessing and post-hoc learning stages that allow behavior-specific dynamics to be dissociated from behaviorally relevant neural dynamics (see section 3.3).

We validate BRAID in multiple simulated datasets with distinct nonlinear structures and show its capability to accurately learn the underlying nonlinear model, resulting in an interpretable representation of intrinsic dynamics. We then apply our method to electrophysiological data recorded from a non-human-primate (NHP) performing sequential reaches (O'Doherty et al., 2017). Our results indicate that accounting for both nonlinearity and sensory inputs improves neural-behavioral prediction, suggesting a more accurate representation of intrinsic behaviorally relevant neural dynamics.

## 2 RELATED WORK

Our work addresses multiple problems simultaneously, which makes it related to various methods that tackle a subset of these problems. A summary of related methods is provided in table 1.

First, a key ability of BRAID is to incorporate measured inputs to disentangle intrinsic dynamics from input dynamics. Other nonlinear modeling methods (Gao et al., 2016; Sussillo et al., 2016; Wu et al., 2017; Pandarinath et al., 2018; Rutten et al., 2020; Hurwitz et al., 2021; Kim et al., 2021; Abbaspourazad et al., 2024; Sani et al., 2024) have not addressed modeling measured external inputs and their impact on neural-behavioral data. As demonstrated by Vahidi et al. (2024), not considering external inputs can lead to the dynamics of these inputs being misinterpreted as intrinsic neural dynamics. To overcome this challenge, Vahidi et al. (2024) introduce a linear dynamical modeling method, termed IPSID, which explicitly incorporates measured external inputs into the model. However, IPSID is an analytical, and strictly linear method that cannot capture nonlinearities. By incorporating the strengths of this linear modeling work into BRAID, we can account for measured external inputs and dissociate their dynamics while allowing every model element to be nonlinear. We use IPSID as a key baseline to show the benefit of enabling nonlinearity in our method (see appendix A.2 for details). We also show the results for the special case of setting all model elements as linear in our method (referred to as linear BRAID), which fits in a linear model similar to IPSID.

Second, a key capability of BRAID is that it dissociates behaviorally relevant neural dynamics into a distinct part of the latent states and prioritizes their learning, while also being able to learn neural-specific and behavior-specific dynamics using additional latent states. Among prior works, two recent nonlinear methods termed DPAD (Sani et al., 2024) and TNDM (Hurwitz et al., 2021) aim to dissociate behaviorally relevant dynamics from other neural dynamics, but neither method dissociates the third category of dynamics, i.e., the behavior-specific dynamics. More importantly, neither

DPAD nor TNDM incorporates external inputs into the model to dissociate intrinsic dynamics from input dynamics. Finally, DPAD learns models based on 1-step-ahead prediction of neural-behavioral data and does not explicitly learn the intrinsic dynamics, whereas BRAID adds $m$-step-ahead predictions into the loss to optimize forecasting and also explicitly learns a generative representation of intrinsic dynamics. TNDM on the other hand is a sequential autoencoder (similar to LFADS, Pandarinath et al., 2018), i.e., it optimizes reconstruction of a window of data after ingesting the entire window as input. We compared our results with both DPAD and TNDM, although DPAD's architecture is closer to ours. In fact, the comparisons with DPAD can also be thought of as ablation studies that show the benefit of incorporating external inputs and forecasting in our method.

Third, we learn behaviorally relevant neural dynamics, or in other words the shared neural-behavioral dynamics, in an initial optimization focused on learning these dynamics, while leaving the learning of other neural dynamics to a separate subsequent optimization. This approach, prioritizes behaviorally relevant neural dynamics in the sense that we can fit models with low dimensional latent states that are purely focused on these dynamics (Sani et al., 2021). Besides DPAD (Sani et al., 2024), a few other works, including TNDM, propose nonlinear approaches for learning dynamics shared between two modalities (Hurwitz et al., 2021; Kramer et al., 2022; Gondur et al., 2024). However, these works use a combined loss to optimize the reconstruction of both modalities in the same optimization. While this approach can capture the dynamics shared between modalities, it does not prioritize them over dynamics specific to either modality (Sani et al., 2024). Moreover, most multi-modal approaches do not model the effect of external inputs (Hurwitz et al., 2021; Gondur et al., 2024). One multi-modal model, termed mmPLRNN (Kramer et al., 2022), which models dynamics of two modalities with a piecewise-linear RNN (see appendix A.2 for details), supports modeling the effect of external inputs, although this capability was not demonstrated in Kramer et al. (2022). Nevertheless, we include comparisons with mmPLRNN with input as one baseline. Finally, as another ablation study to assess the importance of prioritizing behaviorally relevant dynamics, we also implement an unsupervised version of BRAID, termed U-BRAID, that removes the behaviorally relevant optimization step and instead learns all neural dynamics in one optimization step while still incorporating external inputs into the model (see section 3 and appendix A.2 for details).

Most other prior nonlinear methods only consider neural signals during modeling without considering behavior or external inputs (Gao et al., 2016; Pandarinath et al., 2018; Hernandez et al., 2020; Rutten et al., 2020; Kim et al., 2021) or do not use dynamic models (Zhou & Wei, 2020; Schneider et al., 2023) and thus are vastly different from our method. Nevertheless, we include comparisons with LFADS (Pandarinath et al., 2018) and CEBRA (Schneider et al., 2023) as additional baselines. We list the differences of some of these methods with our method in table 1.

Table 1: Related works (see Discussion). ELBO: evidence lower bound, LL: log-likelihood.

| Method | Nonlinear | Prioritize behaviorally relevant | Dissociate non-neural | Dissociate intrinsic | Training objective |
|---|---|---|---|---|---|
| IPSID | ✗ | ✓ | ✓ | ✓ | Projection-based |
| TNDM | ✓ | ✓ | ✗ | ✗ | Multi-modal ELBO |
| LFADS | ✓ | ✗ | ✗ | ✗ | ELBO |
| mmPLRNN | ✓ | ✗ | ✗ | ✓ | Multi-modal ELBO |
| DPAD | ✓ | ✓ | ✗ | ✗ | 1-step-ahead LL |
| CEBRA | ✓ | ✓ | ✗ | ✗ | Contrastive |
| **BRAID** | ✓ | ✓ | ✓ | ✓ | $m$-step-ahead LL |

## 3 METHODS

### 3.1 BRAID MODEL

We model the neural activity ($\mathbf{y}_k \in \mathbb{R}^{n_y}$) and behavior ($\mathbf{z}_k \in \mathbb{R}^{n_z}$) as observations of a nonlinear dynamical system that is driven by measured ($\mathbf{u}_k \in \mathbb{R}^{n_u}$) and/or unmeasured inputs ($\mathbf{w}_k \in \mathbb{R}^{n_x}$):

$$\begin{cases} \mathbf{x}_{k+1}^s &= \boldsymbol{A}_{fw}(\mathbf{x}_k^s) + \boldsymbol{K}_{fw}(\mathbf{u}_k) + \mathbf{w}_k \\ \mathbf{y}_k &= \boldsymbol{C}_{\mathbf{y}}(\mathbf{x}_k^s, \mathbf{u}_k) + \mathbf{v}_k \\ \mathbf{z}_k &= \boldsymbol{C}_{\mathbf{z}}(\mathbf{x}_k^s, \mathbf{u}_k) + \boldsymbol{\epsilon}_k \end{cases} \quad (1)$$

Here, $\mathbf{x}_k^s \in \mathbb{R}^{n_x}$ represent the latent states of the system and evolve according to intrinsic dynamics $\boldsymbol{A}_{fw}$. $\mathbf{v}_k \in \mathbb{R}^{n_y}$ and $\boldsymbol{\epsilon}_k \in \mathbb{R}^{n_z}$ represent observation noises. Given this dynamical system, one can recursively infer the latent state from neural observations $\mathbf{y}_k$ using an RNN as follows

$$\mathbf{x}_{k+1|k} \;\; = \;\; \boldsymbol{A}(\mathbf{x}_{k|k-1}) + \boldsymbol{K}(\mathbf{y}_k, \mathbf{u}_k) \tag{2}$$

where $\mathbf{x}_{k+1|k}$ (or simply $\mathbf{x}_{k+1}$) is defined as the inferred latent state based on $\{\mathbf{y}_1, ..., \mathbf{y}_k\}$ and $\{\mathbf{u}_1, ..., \mathbf{u}_k\}$. Given the latent nature of the states, even when inference is optimal (e.g., in a Kalman filter), the inferred states will *not* be equal to the internal states $\mathbf{x}_k^s$ in equation 1 (Katayama, 2006), which is why we use different notations for the states in equations 1 and 2. More importantly, note that $\boldsymbol{A}$ and $\boldsymbol{K}$ in equation 2 are distinct from $\boldsymbol{A}_{fw}$ and $\boldsymbol{K}_{fw}$ in equation 1. This is because $\boldsymbol{A}$ and $\boldsymbol{K}$ represent the "predictor form" representation of dynamics, describing how the inferred latent state recursively evolves over time as samples of $\mathbf{y}_k$ and $\mathbf{u}_k$ are observed, whereas $\boldsymbol{A}_{fw}$ and $\boldsymbol{K}_{fw}$ represent the "generative form" representation of dynamics that describe how the latent states themselves evolve, purely based on their *intrinsic dynamics* – so $\boldsymbol{A}_{fw}$ is what ultimately describes the intrinsic dynamics. For linear systems, there is an analytical bidirectional relationship between predictor and generative form representations (defined by the Kalman filter, see Katayama, 2006), whereas for nonlinear systems in general, this relationship is not known. Thus, we devise an approach that allows us to learn both representations of dynamics from data.

Critically, to predict the latent state (or neural-behavioral data) multiple ($m > 1$) steps into the future without new neural observations and using only new inputs $\mathbf{u}_k$, we need to propagate the latent state ahead according to its *intrinsic* dynamics, i.e., the "generative form" representation of dynamics, as

$$\mathbf{x}_{k+m|k} \;\; = \;\; \boldsymbol{A}_{fw}(\mathbf{x}_{k+m-1|k}) + \boldsymbol{K}_{fw}(\mathbf{u}_{k+m-1}) \tag{3}$$

where $\mathbf{x}_{k+m|k}$ denotes the latent state at time step $k + m$, generated given $\{\mathbf{y}_1, ..., \mathbf{y}_k\}$ and $\{\mathbf{u}_1, ..., \mathbf{u}_{k+m-1}\}$. Note that for $m = 2$, the right hand side of equation 3 would have $\mathbf{x}_{k+1|k}$, which is given by equation 2. Thus, $m$-step-ahead inference of the latent state engages both the predictor and generative form representations of the dynamics via equations 2 and 3, respectively. As an alternative interpretation, the $m$-step-ahead prediction of the latent state (or neural or behavioral data), for $m > 1$, involves two RNNs operating in complementary fashion (figure 1b):

1. The first RNN (*RNN*, parameterized by $\boldsymbol{A}$ and $\boldsymbol{K}$) takes in neural and input time series and recursively estimates the 1-step-ahead prediction ($\mathbf{x}_{k|k-1}$).

2. The second RNN (*RNN$_{fw}$*, parameterized by $\boldsymbol{A}_{fw}$ and $\boldsymbol{K}_{fw}$) takes in the 1-step-ahead predicted state from the first RNN and propagates it $m-1$ additional steps ahead according to the intrinsic latent dynamics of the model, to get the $m$-step-ahead predictions ($\mathbf{x}_{k+m-1|k-1}$, for $m > 1$).

Overall, the BRAID model is comprised of six distinct transformations: $\boldsymbol{A}(\cdot)$, $\boldsymbol{A}_{fw}(\cdot)$, $\boldsymbol{K}(\cdot)$, $\boldsymbol{K}_{fw}(\cdot)$, $\boldsymbol{C}_{\mathbf{z}}(\cdot)$, and $\boldsymbol{C}_{\mathbf{y}}(\cdot)$. $\boldsymbol{A}/\boldsymbol{A}_{fw}$ describe predictor/generative form recursions of the latent state. $\boldsymbol{K}/\boldsymbol{K}_{fw}$ describe predictor/generative form encoders. $\boldsymbol{C}_{\mathbf{z}}$ and $\boldsymbol{C}_{\mathbf{y}}$ describe behavior and neural decoders. We implement these six transformations as multi-layer perceptrons (MLPs) with arbitrary user-specified number of units and hidden layers. As a special case, any (or all) of these mappings can be replaced by a linear mapping (i.e., an MLP with no hidden layer and a linear activation).

We learn the parameters specifying all six transformations of the model by optimizing a weighted sum of $m$-step-ahead neural-behavioral prediction errors (for $m \in [m_1, m_2, \ldots, m_L]$) as our losses

$$
\begin{aligned}
L_{\mathbf{z}} = \sum_{i=1}^{L} \alpha_{z_{m_i}} \mathrm{MSE}(\mathbf{z}_{k+m_i}, \boldsymbol{C}_{\mathbf{z}}(\mathbf{x}_{k+m_i|k}, \mathbf{u}_{k+m_i})) \\
L_{\mathbf{y}} = \sum_{i=1}^{L} \alpha_{y_{m_i}} \mathrm{MSE}(\mathbf{y}_{k+m_i}, \boldsymbol{C}_{\mathbf{y}}(\mathbf{x}_{k+m_i|k}, \mathbf{u}_{k+m_i}))
\end{aligned}
\tag{4}
$$

where $\mathrm{MSE}(\cdot)$ indicates the mean-squared error loss, $L$ denotes the number of steps ahead simultaneously included in the loss, and $\alpha_{z_{m_i}}$ and $\alpha_{y_{m_i}}$ denote the weights used in the sum. In this work, we always set $\alpha_{z_{m_i}}$ and $\alpha_{y_{m_i}}$ to 1. Moreover, although the decoders in BRAID can optionally take both the latent state and the external input $\mathbf{u}_k$ (lines 2-3 of equation 1), in our real data analyses we do not provide $\mathbf{u}_k$ to decoders and generate predictions only based on the latent states.

## 3.2 Prioritization of behaviorally relevant over other neural dynamics

To dissociate behaviorally relevant neural dynamics from other neural dynamics and prioritize the former, we break the latent state $\mathbf{x}_k$ into two sections ($\mathbf{x}_k^{(1)}$ and $\mathbf{x}_k^{(2)}$) and learn these two sections

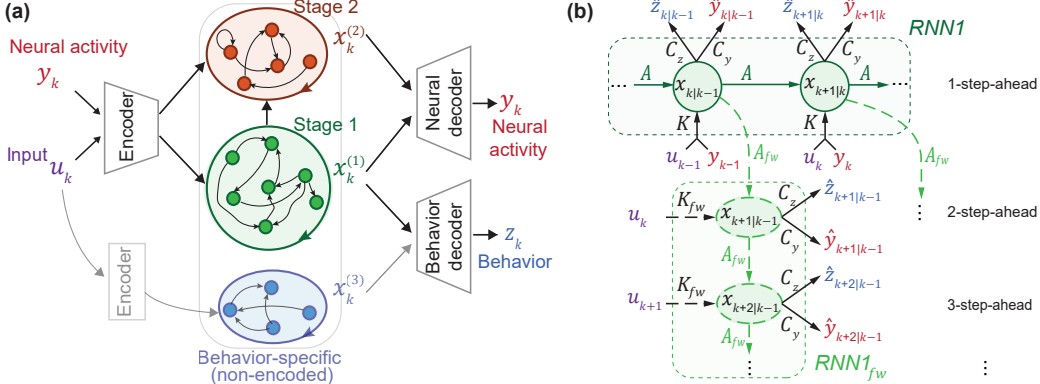

Figure 1: **BRAID model architecture**. **(a)** BRAID dissociates the dynamics of neural-behavioral data into three latent states $\mathbf{x}_k^{(1)}$, $\mathbf{x}_k^{(2)}$, and $\mathbf{x}_k^{(3)}$: 1) the dynamics shared between neural and behavioral modalities (learned in stage 1 by *RNN1* and *RNN1$_{fw}$*), 2) any remaining dynamics private to neural activity (learned in stage 2 by *RNN2* and *RNN2$_{fw}$*), and 3) Input-driven, behavior-specific dynamics not encoded in neural activity (learned per section 3.3 by *RNN3* and *RNN3$_{fw}$*). **(b)** For each latent state, we simultaneously learn a predictor and generative form representation of the dynamics (denoted by $\boldsymbol{A}$ and $\boldsymbol{A}_{fw}$), by optimizing $m$-step-ahead prediction of neural-behavioral data. This interconnected two-RNN system is visualized for *RNN1* and *RNN1$_{fw}$* in this computation graph. The superscript $.^{(1)}$ indicating that parameters are for *RNN1* and *RNN1$_{fw}$* is omitted for simplicity.

in two learning stages. We denote the model parameters associated with each model section using a $.^{(1)}$ or $.^{(2)}$ superscript, e.g., $\boldsymbol{A}^{(1)}$ and $\boldsymbol{A}^{(2)}$. We provide the full two-section formulation for the model in appendix A.1.1 and the optimization details in appendix A.1.2. Briefly, each of the two learning stages consist of 2 optimizations, as follows:

**Stage 1**: Learning *RNN1* and *RNN1$_{fw}$*

1a Learn $\boldsymbol{A}^{(1)}$, $\boldsymbol{A}_{fw}^{(1)}$, $\boldsymbol{K}^{(1)}$, $\boldsymbol{K}_{fw}^{(1)}$, and $\boldsymbol{C}_{\mathbf{z}}^{(1)}$, and extract latent states $\mathbf{x}_k^{(1)}$ and $\mathbf{x}_{k+m|k}^{(1)}$ (for $m > 1$) by minimizing the behavior prediction loss $L_{\mathbf{z}}$ from equation 4.

1b Learn $\boldsymbol{C}_{\mathbf{y}}^{(1)}$ by predicting neural data from $\mathbf{x}_k^{(1)}$ and $\mathbf{x}_{k+m|k}^{(1)}$, while minimizing the neural prediction loss $L_{\mathbf{y}}$ from equation 4.

**Stage 2**: Learning *RNN2* and *RNN2$_{fw}$*

2a Learn $\boldsymbol{A}^{(2)}$, $\boldsymbol{A}_{fw}^{(2)}$, $\boldsymbol{K}^{(2)}$, $\boldsymbol{K}_{fw}^{(2)}$, and $\boldsymbol{C}_{\mathbf{y}}^{(2)}$, and extract latent states $\mathbf{x}_k^{(2)}$ and $\mathbf{x}_{k+m|k}^{(2)}$ (for $m > 1$) by minimizing the neural loss $L_{\mathbf{y}}$, while including outputs of stage 1b as part of the predictions.

2b Learn $\boldsymbol{C}_{\mathbf{z}}^{(2)}$ by predicting behavior from $\mathbf{x}_k^{(2)}$ and $\mathbf{x}_{k+m|k}^{(2)}$, while minimizing the behavior loss $L_{\mathbf{z}}$ from equation 4, and including outputs of stage 1a as part of the predictions.

The explicit dissociation of the relevant dynamics and the above two-stage optimization allow us to first preferentially learn the (low-dimensional) shared dynamics between the two observations, i.e., the behaviorally relevant neural dynamics ($\mathbf{x}_k^{(1)}$), in stage 1. Then in stage 2, we learn any residual neural dynamics ($\mathbf{x}_k^{(2)}$), which, as depicted in figure 1a, can depend on the behaviorally relevant dynamic (see appendix A.1 for details). The optional stage 2 is of interest for explaining neural dynamics beyond the ones related to behavior. This multi-stage approach has similarities to Sani et al. (2024), but here we have: 1) additional signals ($\mathbf{u}_k$), 2) different losses, 3) a forecasting RNN within each model section (figure 1b), and additional steps that are discussed in the next section.

### 3.3 DISSOCIATION OF BEHAVIOR-SPECIFIC DYNAMICS

Optimizing behavior prediction (stage 1a) given neural activity $\mathbf{y}_k$ and input $\mathbf{u}_k$ can lead to learning behavior dynamics that are predictable from the input but are not encoded in the recorded neural activity. Although learning such behavior-specific dynamics enhances behavior decoding, it poses an interpretation challenge for neuroscience applications because one would not know what part of

the learned dynamics are represented in the recorded brain regions. As shown in Vahidi et al. (2024), this may lead to a misinterpretation of input-driven behavior-specific dynamics as intrinsic dynamics of the recorded brain region. To mitigate this possibility, we develop two additional steps in our method, that can 1) exclude such behavior-specific dynamics from $\mathbf{x}_k^{(1)}$, and 2) learn them separately as a distinct latent state $\mathbf{x}_k^{(3)}$ (figure 1a). Details are provided in appendix A.1.3. Briefly, *first*, to exclude behavior-specific dynamics, we introduce an optional preprocessing stage that predicts behavior from neural data, and passes this neurally-predicted behavior to be used in stages 1a and 2b. This preprocessing step ensures that the behaviorally relevant states learned in stage 1 ($\mathbf{x}_k^{(1)}$) are encoded in recorded neural activity $\mathbf{y}_k$, which can be crucial for interpretability in neuroscience studies. In our analyses of the real datasets (section 4.2), we always include this preprocessing step. *Second*, to still be able to learn behavior-specific dynamics, we add an optional post-hoc learning step (i.e., stage 3) that fits *RNN3* and *RNN3*$_{fw}$ to any unexplained behavior and learns these input-driven behavior-specific dynamics as a distinct latent state $\mathbf{x}_k^{(3)}$. As we show in simulations (see section A.5.1 and figure A.2), the preprocessing step can exclude non-encoded behavior dynamics. When desired in an application, the optional stage 3 can learn such dynamics to offset any behavior decoding loss incurred due to the preprocessing step, while still maintaining the interpretability of the model. We did not apply this post-hoc step in our real data analyses (section 4.2).

## 3.4 INFERENCE AND EVALUATION METRICS

After learning BRAID's parameters, we can readily use the learned mappings $\boldsymbol{A}$, $\boldsymbol{K}$ (and $\boldsymbol{A}_{fw}$ and $\boldsymbol{K}_{fw}$) to infer the 1-(and multi)-step-ahead predicted states $\mathbf{x}_k$ (and $\mathbf{x}_{k+m|k}$) using equations 2 (and 3) for the held-out test data. Predicted neural activity and behavior are obtained by applying their corresponding decoders $\boldsymbol{C}_y$ and $\boldsymbol{C}_z$ to these inferred states. We also use the term "decoding" for behavior predictions because our model predicts behavior only using neural data and inputs, and never using behavior itself. To evaluate the performance of our models, we perform 5- and 2-fold cross-validation, for real data and simulation analyses, respectively. We report Pearson's Correlation Coefficient (CC) and in some cases (see table A.8) also the coefficient of determination ($R^2$) between the predicted and actual observation, averaged over dimensions. We further report the $m$-step-ahead prediction accuracy, which reflects how well the intrinsic dynamics $\boldsymbol{A}_{fw}$ are learned. For simulation analyses with linear recursions ($\boldsymbol{A}_{fw}$), we additionally evaluate the learned intrinsic dynamics by comparing the eigenvalues of $\boldsymbol{A}_{fw}$ between the true and learned model (see appendix A.1.5).

## 4 EXPERIMENTAL RESULTS

### 4.1 SIMULATION EXPERIMENTS

We validated BRAID in three simulations with different nonlinear neural-behavioral-input structures to show that it can learn intrinsic behaviorally relevant neural dynamics in presence of inputs.

### 4.1.1 BRAID ACHIEVES NEAR OPTIMAL NEURAL-BEHAVIORAL PREDICTIVE ACCURACY IN NONLINEAR INPUT-DRIVEN SIMULATIONS

First, we considered an input-driven dynamical system as in equation 1, but with only the behavior mapping $\boldsymbol{C}_\mathbf{z}$ being nonlinear with the mapping $f_{C_z}(\nu) := a\sin(\nu) + b\nu$, as detailed in section A.4.2 (figure 2a). We generated 10 random parameter sets as our true models and generated data from them. First, we implemented an automatic selection of nonlinearity for BRAID by setting each of $\boldsymbol{A}$, $\boldsymbol{K}$, $\boldsymbol{C}_\mathbf{y}$, or $\boldsymbol{C}_\mathbf{z}$ to linear or nonlinear, resulting in $2^4$ different BRAID models, and finding the model with the best behavior decoding in the training data. Across all 10 realizations and 2 cross-validated folds, setting the behavior decoder $\boldsymbol{C}_\mathbf{z}$ to be nonlinear was correctly identified as the best performing nonlinearity in 100% of the cases. Additionally, we evaluated BRAID with nonlinearity only in one of $\boldsymbol{A}$, $\boldsymbol{K}$, $\boldsymbol{C}_\mathbf{y}$, or $\boldsymbol{C}_\mathbf{z}$. The model with nonlinear behavior decoder $\boldsymbol{C}_\mathbf{z}$ outperformed other nonlinearity choices as well as linear models (i.e., IPSID and the fully linear BRAID) in behavior decoding and neural prediction. BRAID further outperformed DPAD, which is nonlinear but does not account for the input $u_k$, in neural-behavioral prediction. In fact, both BRAID with nonlinear $\boldsymbol{C}_\mathbf{z}$ and BRAID with automatic nonlinearity selection achieved almost the same neural-behavioral

prediction as the true simulated models, demonstrating BRAID's success in accurately learning the nonlinear input-driven dynamical system (table 2).

Table 2: 1-step-ahead prediction results for nonlinear simulations with sinusoidal behavior mapping. Mean $\pm$ s.e.m. is across 20 runs (10 datasets, 2 folds). State dimension is always set to ground truth. True model's outcome indicates the "Ideal" accuracy. **Bold**: within 1 s.e.m. of ideal.

| Method | Behavior decoding CC | Neural prediction CC |
|---|---|---|
| IPSID | $0.4567 \pm 0.0527$ | $\mathbf{0.8901 \pm 0.0304}$ |
| linear BRAID | $0.4558 \pm 0.0528$ | $\mathbf{0.8893 \pm 0.0304}$ |
| DPAD *Nonlin* $C_\mathbf{z}$ | $0.4958 \pm 0.0620$ | $0.3767 \pm 0.0680$ |
| BRAID *Nonlin* $\mathbf{A}$ | $0.4735 \pm 0.0572$ | $0.8127 \pm 0.0528$ |
| BRAID *Nonlin* $\mathbf{K}$ | $0.7680 \pm 0.0467$ | $0.6983 \pm 0.0473$ |
| BRAID *Nonlin* $C_\mathbf{y}$ | $0.4558 \pm 0.0528$ | $\mathbf{0.8887 \pm 0.0305}$ |
| BRAID *Nonlin* $C_\mathbf{z}$ | $\mathbf{0.8696 \pm 0.0487}$ | $\mathbf{0.8913 \pm 0.0305}$ |
| BRAID *Auto Nonlin* | $\mathbf{0.8693 \pm 0.0487}$ | $\mathbf{0.8913 \pm 0.0305}$ |
| True model (ideal) | $0.8737 \pm 0.0486$ | $0.8921 \pm 0.0306$ |

### 4.1.2 BRAID DISSOCIATES INTRINSIC DYNAMICS FROM INPUT DYNAMICS IN SIMULATIONS

Next, we sought to validate BRAID's ability to disentangle intrinsic and input-driven contributions to neural dynamics. BRAID simultaneously learns a predictor form and a generative form representation of the dynamics, the latter of which directly describes the intrinsic dynamics in terms of the mapping $\mathbf{A}_{fw}(\cdot)$ (section 3.1, figure 1b). In this simulation, we kept the ground truth state transitions linear so that we could precisely quantify the intrinsic dynamics and their learning error via the eigenvalues of the state transition matrix $\mathbf{A}_{fw}$ (appendix A.1.5). We analyzed data from three sets of simulated dynamical systems with distinct nonlinear structures (see appendix A.4 for details): (1) Spiral behavior manifold (figure 2a-d), (2) trigonometric behavior manifold (also explained in section 4.1.1, figure 2e-h), and (3) trigonometric input-encoder (figure A.1). For each simulation, we generated realizations from 10 different systems with randomly generated sets of parameters. Across all three simulations, BRAID accurately learned the intrinsic dynamics, resulting in smaller error in eigenvalues of the transition matrix compared with DPAD, which is nonlinear but does not consider the input, and compared with linear BRAID and IPSID, which consider input but are linear (figures 2c,g, A.1d). We demonstrate in an ablation study that learning a separate generative model for forecasting is crucial for correct learning of the intrinsic dynamics (table A.3). These more accurate intrinsic dynamics coupled with the input also resulted in BRAID achieving better behavior decoding (figures 2b,f , A.1a) and neural prediction (figure A.1c) compared to baselines. These results suggest that failing to account for either nonlinearity or input may lead to less accurate models and a misinterpretation of the intrinsic dynamics in nonlinear data.

Another metric for how well intrinsic dynamics are learned is forecasting, where behavior is predicted multiple steps into the future, without observing new neural data and only by observing the future input (section 3.1). Forecasting evolves the state dynamics according to the learned intrinsic dynamics (equation 3) and thus validates their accurate learning. We performed forecasting up to 32 steps ahead and found that the nonlinear models with input consistently outperformed linear models as well as the nonlinear DPAD, which does not consider input (figures 2d,h and A.1e,f).

### 4.2 NON-HUMAN PRIMATE MOTOR CORTICAL ACTIVITY DURING REACHING

We applied our method to a publicly available dataset recorded from a non-human primate (NHP) performing reaching movements O'Doherty et al. (2017) (figure 3a). We took either the smoothed spike counts or raw LFP from primary motor cortex (M1) as neural time-series $\mathbf{y}_k$, fingertip's position and velocity as behavior $\mathbf{z}_k$, and sensory task instructions (target location) as the input $\mathbf{u}_k$ (appendix A.3). Sensory inputs can have their own dynamics, which are distinct from the intrinsic dynamics of the motor cortex. Our goal is to learn the intrinsic dynamics in M1 related to movement while disentangling them from the dynamics of sensory input and also from any behavior-specific dynamics. Therefore, we importantly include BRAID's behavior preprocessing stage (section 3.3).

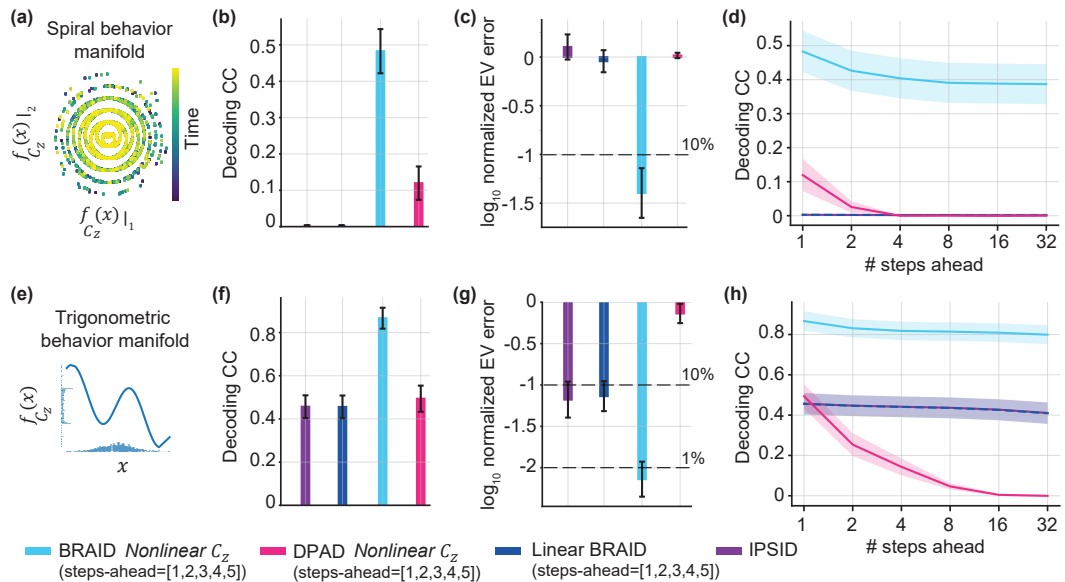

Figure 2: **BRAID better learns the intrinsic shared dynamics by simultaneously modeling input and nonlinearity, and by optimizing forecasting**. **(a-d)** Results for simulations with a spiral behavioral manifold. **(b)** 1-step-ahead behavior decoding for nonlinear BRAID, nonlinear DPAD, linear BRAID, and IPSID **(c)**. Error in identifying intrinsic dynamics of the true model quantified by the error in learning the eigenvalues of $\boldsymbol{A}_{fw}$. **(d)** Behavior decoding forecasts for 1 to 32 steps ahead, enabled by learning the intrinsic dynamics ($\boldsymbol{A}_{fw}$), with predictions optimized for $[1, 2, 3, 4, 5]$-steps-ahead (section 3). **(e-h)** Same as (a-d) for simulations with a trigonometric behavior mapping.

We fitted BRAID with different nonlinearity choices as in our simulations: (1) nonlinear recursion $\boldsymbol{A}(\cdot)/\boldsymbol{A}_{fw}(\cdot)$, (2) nonlinear encoder $\boldsymbol{K}(\cdot)/\boldsymbol{K}_{fw}(\cdot)$, (3) nonlinear decoders $\boldsymbol{C}_{\mathbf{y}}(\cdot)$ and $\boldsymbol{C}_{\mathbf{z}}(\cdot)$, and a fully linear variant, linear BRAID. We included $m = [1, 2, 4, 8]$-steps-ahead predictions in the BRAID loss, which for $m > 1$ engage the intrinsic behaviorally relevant dynamics ($\boldsymbol{A}_{fw}$) and allow their learning. To evaluate the learned intrinsic dynamics, we report neural-behavioral forecasting accuracy for different step-ahead horizons (figures 3b-c, and A.3 for LFP), while tabulating the 4-steps-ahead (i.e., 200ms) results to highlight BRAID's advantage in forecasting (tables 3 and A.4). In addition to these results in the low-dimensional regime (stage 1 only, $n_x = n_1 = 16$), we also report the results in the high-dimensional regime (both stages, $n_x = 64$, $n_1 = 16$) (table 3, figures A.5 and A.6). Among nonlinearity configurations, BRAID with nonlinear decoders provided the best fit to neural-behavioral data (table A.4). Moreover, BRAID's behavior forecasting performance improved as more neurons were included (table A.7).

Next, we compared BRAID's neural-behavioral forecasting to several ablation baselines (table 3 and figures 3 and A.5). First, BRAID outperformed linear BRAID, i.e., a similar but fully linear model. Second, BRAID outperformed DPAD (Sani et al., 2024), which can have decoder nonlinearities but does not consider inputs. BRAID's advantage shows the importance of considering the effects of sensory inputs on neural-behavioral dynamics. Third, we compared to U-BRAID, which removes the first stage of BRAID and thus loses prioritization. BRAID consistently outperformed U-BRAID in behavioral forecasting, but did not match the neural forecasting of U-BRAID unless it was given enough latent state dimensions (table 3, $n_x = 64$), which is expected given U-BRAID's singular objective being neural prediction. This comparison shows the benefit of prioritization for learning low-dimensional representations of intrinsic behaviorally relevant dynamics, and confirms that with its stage 2, BRAID can capture any remaining non-behavioral neural dynamics. Finally, BRAID's low dimensional latent state trajectories were better separated for different movement directions compared to those of U-BRAID and DPAD, suggesting that BRAID's latent states are more congruent with behavior, i.e., more behaviorally relevant (figure A.4).

We also compared BRAID's neural-behavioral forecasting performance to mmPLRNN, which models multi-modal data using piecewise-linear RNNs, and has the option to model inputs although prior

Table 3: Forecasting performance (4-step-ahead) compared to baselines in NHP dataset for models with low ($n_x = 16$) and high-dimensional ($n_x = 64$) latent states. $n_1 = 16$ for BRAID, linear BRAID, and DPAD. $R^2$ results were similar (table A.8). Tables A.4, A.5 and A.6 have additional results.

| Method | Behavior forecasting CC | | Neural forecasting CC | |
|---|---|---|---|---|
| | $n_x = 16$ | $n_x = 64$ | $n_x = 16$ | $n_x = 64$ |
| linear BRAID | $0.7453 \pm 0.0066$ | $0.7409 \pm 0.0059$ | $0.1767 \pm 0.0054$ | $0.3784 \pm 0.0078$ |
| DPAD | $0.6706 \pm 0.0096$ | $0.7352 \pm 0.0079$ | $0.2067 \pm 0.0062$ | $0.3611 \pm 0.0080$ |
| U-BRAID | $0.7663 \pm 0.0069$ | $\mathbf{0.8049 \pm 0.0068}$ | $\mathbf{0.4089 \pm 0.0076}$ | $\mathbf{0.4185 \pm 0.0074}$ |
| mmPLRNN | $0.6851 \pm 0.0143$ | $0.7328 \pm 0.00361$ | $0.3162 \pm 0.0107$ | $0.3570 \pm 0.0223$ |
| BRAID (ours) | $\mathbf{0.8042 \pm 0.0085}$ | $0.7970 \pm 0.0086$ | $0.3274 \pm 0.0078$ | $\mathbf{0.4123 \pm 0.0077}$ |

work had not explored this input option. BRAID outperformed input-driven mmPLRNN networks in forecasting both behavior and neural activity, across all forecasting horizons, with the exception of 2-steps-ahead neural prediction (table 3, figure 3). Note that BRAID's better decoding is achieved despite the fact that mmPLRNN, by design, incorporates behavior *as an input* during inference, whereas BRAID and DPAD do not. Also, we compared BRAID to TNDM (Hurwitz et al., 2021), a nonlinear sequential autoencoder that models neural-behavioral dynamics, but does not include the effect of external inputs. We extended TNDM beyond the original work to create a version that also adds sensory inputs as an additional input besides neural activity (appendix A.2.6). We analyzed non-smoothed spike counts from the same dataset and found that BRAID significantly outperformed TNDM in behavior decoding while achieving comparable neural prediction (table A.5). Extending TNDM to include inputs significantly improved its behavior decoding, but it still did not reach that of BRAID (table A.5). Finally, we compared BRAID with CEBRA (Schneider et al., 2023), which is a non-dynamic convolutional encoder with a contrastive loss on behavior (section A.2.8), and with LFADS (Pandarinath et al., 2018), which is an unsupervised sequential autoencoder (section A.2.7). BRAID outperformed both these baselines in neural-behavioral prediction (table A.6).

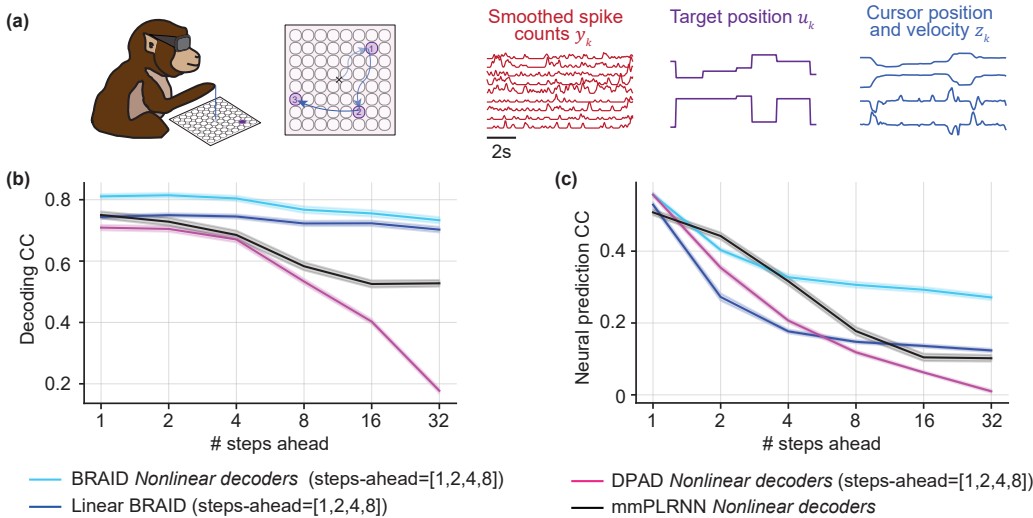

Figure 3: **BRAID outperforms baselines in neural-behavioral forecasting**. **(a)** Dataset and task visualization. **(b, c)** Behavior and neural activity forecasting correlation coefficient (CC) for BRAID, linear BRAID, DPAD, and mmPLRNN at low state dimension regime ($n_x$=16). Shaded areas show the s.e.m. across the 7 recording sessions and 5 cross-validation folds.

# 5 DISCUSSION

We introduced BRAID, a method for input-driven nonlinear dynamical modeling that disentangles intrinsic shared dynamics between two observation modalities from the effect of input. Here, we assume some external inputs are measured (e.g., from sensory stimuli or other brain regions) and

are available for modeling. This approach is distinct from the input-inference approach, where unmeasured inputs are inferred from measured neural activity (Pandarinath et al., 2018; Schimel et al., 2022). These two approaches are in a sense complementary. In our approach, any measured inputs can be explicitly incorporated into the model to dissociate their dynamics from the intrinsic dynamics of the measured neural-behavioral data. Practically, one cannot measure all inputs to a given brain area, so our approach does not rule out the influence of unmeasured inputs on the learned intrinsic dynamics. One could thus use the input-inference approach to infer such unmeasured inputs from all measured signals. Note, however, that inferred inputs are ultimately a function of measured signals and thus do not add any new information (unlike measured inputs); rather they can be thought of as a decomposition of the measured signals based on certain assumptions (e.g., smoothness).

While BRAID's learning is supervised by behavior, it only uses neural activity and input (*not* behavior) during inference. This supervision allows stage 1 to extract behaviorally relevant intrinsic dynamics with priority, while later stages learn neural-specific or behavior-specific dynamics in independent optimizations. This multi-stage learning has similarities to some prior works (Sani et al., 2021; Vahidi et al., 2024; Sani et al., 2024; Oganesian et al., 2024), but is fundamentally different from other works that use a single multi-modal optimization loss (Kramer et al., 2022; Gondur et al., 2024; Hurwitz et al., 2021), which may miss prioritization of shared dynamics over unshared dynamics (Sani et al., 2024). Some of the latter group are indeed focused on multi-modal inference and fuse all shared and unshared information into the same latent space (Kramer et al., 2022; Gondur et al., 2024). Multi-stage methods avoid this fusion by focusing on shared dynamics during a first optimization stage with only cross-modality prediction (e.g., behavior decoding) as the objective.

To disentangle input dynamics from intrinsic dynamic, BRAID consists of three stages, each learning a predictor and a generator model. In each stage, BRAID's predictor model in that stage (e.g., *RNN1*) infers the latent states, which are subsequently employed as the input to compute $m$-step-ahead predictions via the associated generative model (e.g., $RNN1_{fw}$). BRAID's predictor and generative models are learned jointly to maximize the $m$-step-ahead log-likelihood. This has analogies to the encoder-decoder architectures such as those commonly used in variational inference (Kingma & Welling, 2013). Specifically, the predictor and generator RNNs in BRAID have roles similar to those of the encoder and decoder in variational inference, respectively. However, in variational inference, the posterior distribution of the unobserved variables given the data is parametrized and a part of the optimization loss aims to enforce that distribution on the inferred latent variables (Chung et al., 2015; Krishnan et al., 2015; Fraccaro et al., 2016; Luk et al., 2024). In contrast, we do not impose such parametrization on the latent states in BRAID. Developing variational methods with the same multi-section architecture and multi-stage learning as BRAID is an interesting future direction.

Similar to many prior works including some in neuroscience (Abbaspourazad et al., 2024; Sani et al., 2024; Chung et al., 2015; Krishnan et al., 2015; Fraccaro et al., 2016; Luk et al., 2024), BRAID has a causal formulation and performs inference by recursively inferring the next latent state after each new observation. This is distinct from some nonlinear methods in neuroscience (Gao et al., 2016; Pandarinath et al., 2018; Hernandez et al., 2020; Hurwitz et al., 2021; Keshtkaran et al., 2022; Gondur et al., 2024; Karniol-Tambour et al., 2024) that perform inference non-causally in time. Thus, BRAID may be useful for real-time decoding in brain-computer interfaces (Shanechi, 2019).

BRAID, as presented, is designed for single-session settings. Extending it for cross-session generalization is an interesting future direction that can follow approaches used in the literature, such as aligning neural manifolds across sessions (Farshchian et al., 2019), adaptive modeling approaches (Ahmadipour et al., 2021; Yang et al., 2021a), or incorporating session-specific read-in and read-out mappings while keeping shared model parameters fixed (Pandarinath et al., 2018). The latter allows training across multiple sessions and adapting to new sessions with minimal data by learning session-specific matrices. Furthermore, BRAID's ability to disentangle intrinsic dynamics may also facilitate generalization across tasks with different sensory instructions.

A fundamental challenge for nonlinear latent state models is that many alternative models may explain the data equally well. Thus, evaluating learned dynamics relies on computable quantities from measured signals, primarily $m$-step-ahead neural-behavioral predictions. While our results suggest that in our dataset having nonlinear decoders provides higher performance than other nonlinearities, this might not be case in another dataset. In practice, the optimal nonlinearity can be selected based on the desired metric as done by BRAID. For linear models, all correct latent models have the same eigenvalues (Katayama, 2006), allowing direct evaluation of learned dynamics (figure 2).

## 6 ACKNOWLEDGMENTS

This work was supported, in part, by the following organizations and grants: National Institutes of Health (NIH) R01MH123770, NIH BRAIN Initiative R61MH135407, NIH RF1DA056402, NIH Director's New Innovator Award DP2-MH126378, Foundation for OCD Research (FFOR), the Office of Naval Research (ONR) Young Investigator Program under contract N00014-19-1-2128, and the Army Research Office (ARO) under contract W911NF-16-1-0368 as part of the collaboration between the US DOD, the UK MOD and the UK Engineering and Physical Research Council (EPSRC) under the Multidisciplinary University Research Initiative (MURI). We sincerely thank the Sabes lab at the University of California San Francisco for making the NHP datasets that we used here publicly available.

## 7 AUTHOR CONTRIBUTIONS

P.V., O.G.S., and M.M.S. developed the new BRAID method and wrote the manuscript. P.V. and O.G.S. implemented the code for the method and the analyses. P.V. performed the analyses. M.M.S. supervised the work.

## 8 REPRODUCIBILITY STATEMENT

To ensure the reproducibility of our work, we are sharing the code for BRAID along with a Python notebook demonstrating its usage at `https://github.com/ShanechiLab/BRAID`. We also provide model architecture and training details in appendix A.1.4. Finally, the dataset we used (O'Doherty et al., 2017) is publicly available for anyone interested in reproducing the results reported in section 4.2.

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

# A APPENDIX

## A.1 METHOD DETAILS

### A.1.1 TWO-SECTION FORMULATION

In equations 1, 2, and 3, we combined both latent state sections of our model ($\mathbf{x}_k^{(1)}$ and $\mathbf{x}_k^{(2)}$) for simpler exposition. Here, we present the complete two-section formulation. The predictor form part of the model (equation 2) can be written as follows:

$$
\begin{cases}
\begin{bmatrix} \mathbf{x}_{k+1|k}^{(1)} \\ \mathbf{x}_{k+1|k}^{(2)} \end{bmatrix} &= \begin{bmatrix} \boldsymbol{A}^{(1)}(\mathbf{x}_{k|k-1}^{(1)}) \\ \boldsymbol{A}^{(2)}(\mathbf{x}_{k|k-1}^{(2)}) \end{bmatrix} + \begin{bmatrix} \boldsymbol{K}^{(1)}(\mathbf{y}_k, \mathbf{u}_k) \\ \boldsymbol{K}^{(2)}(\mathbf{y}_k, \mathbf{u}_k, \mathbf{x}_{k|k-1}^{(1)}) \end{bmatrix} \\
\hat{\mathbf{y}}_{k|k-1} &= \boldsymbol{C}_{\mathbf{y}}^{(1)}(\mathbf{x}_{k|k-1}^{(1)}, \mathbf{u}_k) + \boldsymbol{C}_{\mathbf{y}}^{(2)}(\mathbf{x}_{k|k-1}^{(2)}, \mathbf{u}_k) \\
\hat{\mathbf{z}}_{k|k-1} &= \boldsymbol{C}_{\mathbf{z}}^{(1)}(\mathbf{x}_{k|k-1}^{(1)}, \mathbf{u}_k) + \boldsymbol{C}_{\mathbf{z}}^{(2)}(\mathbf{x}_{k|k-1}^{(2)}, \mathbf{u}_k)
\end{cases}
\tag{A.1}
$$

where we have also included the two-section formulation for the prediction of observations as lines 2-3 of the equation. As before, $\mathbf{y}_k \in \mathbb{R}^{n_y}$ and $\mathbf{z}_k \in \mathbb{R}^{n_z}$ are the observed high-dimensional neural activity and behavior respectively while $\mathbf{u}_k \in \mathbb{R}^{n_u}$ represents the measured inputs to the dynamical system. Here, the overall latent state, $\mathbf{x}_k \in \mathbb{R}^{n_x}$, which describes the dynamics underlying the neural-behavioral data, is constructed such that the behaviorally relevant neural dynamics, represented by $\mathbf{x}_k^{(1)} \in \mathbb{R}^{n_1}$, are dissociated from the the irrelevant ones, represented by $\mathbf{x}_k^{(2)} \in \mathbb{R}^{n_x - n_1}$.

The predictor form RNNs in equation A.1 (i.e., *RNN1* and *RNN2*) are complemented by another set of RNNs (i.e., *RNN1$_{fw}$* and *RNN2$_{fw}$*) that constitute the generative form part of the model (equation 3), and enable $m$-step-ahead (for $m > 1$) prediction of latent states and neural-behavioral data. The generative RNNs were again shown with a combined latent state in equation 3 for simpler exposition. The following equations show the complete two-section formulation:

$$
\begin{cases}
\begin{bmatrix} \mathbf{x}_{k+m|k}^{(1)} \\ \mathbf{x}_{k+m|k}^{(2)} \end{bmatrix} &= \begin{bmatrix} \boldsymbol{A}_{fw}^{(1)}(\mathbf{x}_{k+m-1|k}^{(1)}) \\ \boldsymbol{A}_{fw}^{(2)}(\mathbf{x}_{k+m-1|k}^{(2)}) \end{bmatrix} + \begin{bmatrix} \boldsymbol{K}_{fw}^{(1)}(\mathbf{u}_{k+m-1}) \\ \boldsymbol{K}_{fw}^{(2)}(\mathbf{u}_{k+m-1}, \mathbf{x}_{k+m-1|k}^{(1)}) \end{bmatrix} \\
\hat{\mathbf{y}}_{k+m|k} &= \boldsymbol{C}_{\mathbf{y}}^{(1)}(\mathbf{x}_{k+m|k}^{(1)}, \mathbf{u}_{k+m}) + \boldsymbol{C}_{\mathbf{y}}^{(2)}(\mathbf{x}_{k+m|k}^{(2)}, \mathbf{u}_{k+m}) \\
\hat{\mathbf{z}}_{k+m|k} &= \boldsymbol{C}_{\mathbf{z}}^{(1)}(\mathbf{x}_{k+m|k}^{(1)}, \mathbf{u}_{k+m}) + \boldsymbol{C}_{\mathbf{z}}^{(2)}(\mathbf{x}_{k+m|k}^{(2)}, \mathbf{u}_{k+m})
\end{cases}
\tag{A.2}
$$

where $m > 1$, and $\mathbf{x}_{k+1|k}^{(1)}$ and $\mathbf{x}_{k+1|k}^{(2)}$ (i.e., $m = 1$) are taken from equation A.1. A visualization of how the formulations in equations A.1 and A.2 are connected is provided in figure 1. In equation A.2, we have also included the two-section formulation for the prediction of neural-behavioral observations as lines 2-3 of the equation, showing that applying the same decoders $\boldsymbol{C}_{\mathbf{y}}^{(1)}/\boldsymbol{C}_{\mathbf{y}}^{(2)}$ and $\boldsymbol{C}_{\mathbf{z}}^{(1)}/\boldsymbol{C}_{\mathbf{z}}^{(2)}$ as in equation A.1 to the $m$-step-ahead predicted latents $\mathbf{x}_{k+m|k}^{(1)}/\mathbf{x}_{k+m|k}^{(2)}$ gives the $m$-step-ahead predictions of the neural-behavioral data ($\hat{\mathbf{y}}_{k+m|k}$ and $\hat{\mathbf{z}}_{k+m|k}$).

Equations A.1 and A.2 together constitute the two-section formulation of the BRAID model, which consists of 12 transformations in total: $\boldsymbol{A}(\cdot)$, $\boldsymbol{A}_{fw}(\cdot)$, $\boldsymbol{K}(\cdot)$, $\boldsymbol{K}_{fw}(\cdot)$, $\boldsymbol{C}_{\mathbf{z}}(\cdot)$, and $\boldsymbol{C}_{\mathbf{y}}(\cdot)$, each having two sections denoted with the $.^{(1)}$ and $.^{(2)}$ superscripts.

### A.1.2 LEARNING ALGORITHM STEPS

In sections A.1.2-A.1.3, we provide detailed formulations of the optimization stages used in BRAID during learning. For simplicity, we explain the optimizations in terms of 1-step ahead predictions, which involve predictor form parameters of the model. Formulations for $m$-step-ahead predictions, which constitute additional terms in the overall loss (equations 4), are analogous to those provided here, but instead of the predictor form parameters (equation A.1) they engage the generative form parameters (equation A.2).

Note that regardless of what step-ahead predictions were included during training, the learned model can be used to predict the latent state and neural-behavioral data at $m$-steps ahead for any desired $m$ using equations 2 and 3 (or A.1 and A.2) together. For example, in figure 3, only $[1, 2, 4, 8]$ step ahead predictions are included in the optimization loss (equation 4, but we evaluate the learned models with predictions up to 32-steps ahead.

We develop a two-stage optimization algorithm for learning parameters of the two sections of the BRAID model (equation A.1). We note that the following 2 stages are sequential. This means that parameters associated with behaviorally relevant states $(\mathbf{x}_k^{(1)})$ are fully learned with *RNN1*, then if needed, the remaining parameters corresponding to non-relevant dynamics $(\mathbf{x}_k^{(2)})$ can be learned via *RNN2*. In all the optimizations described below, we use the mean-squared-error (MSE) of predicting observations as the loss function, but we note that the MSE is proportional to the negative log-likelihood (NLL) for isotropic Gaussian-distributed data. Below we provide the details for the 4 optimizations that are performed in the two learning stages of BRAID.

**Stage 1**:

**1a** First, BRAID learns a recurrent neural network (*RNN1*) with $n_1$ states, to minimize behavior prediction MSE given past neural data and inputs (equation A.3). This ensures that *RNN1* only learns neural dynamics that are relevant to (i.e., predictive of) behavior. The states of *RNN1* constitute the first set of latent states in the BRAID model: $\mathbf{x}_k^{(1)}$. This optimization step can be formulated as:

$$\begin{cases} \mathbf{x}_{k+1}^{(1)} &= \boldsymbol{A}^{(1)}(\mathbf{x}_k^{(1)}) + \boldsymbol{K}^{(1)}(\mathbf{y}_k, \mathbf{u}_k) \\ \mathbf{z}_k &= \boldsymbol{C}_{\mathbf{z}}^{(1)}(\mathbf{x}_k^{(1)}, \mathbf{u}_k) \\ loss &: MSE(\mathbf{z}_k, \boldsymbol{C}_{\mathbf{z}}^{(1)}(\mathbf{x}_k^{(1)}, \mathbf{u}_k)) \end{cases} . \tag{A.3}$$

**1b** Next, in a second optimization, we learn a transformation $\boldsymbol{C}_{\mathbf{y}}^{(1)}(\cdot)$ that maps $\mathbf{x}_k^{(1)}$ to neural activity while minimizing neural prediction MSE (equation A.4):

$$\begin{cases} \mathbf{y}_k &= \boldsymbol{C}_{\mathbf{y}}^{(1)}(\mathbf{x}_k^{(1)}, \mathbf{u}_k) \\ loss &: MSE(\mathbf{y}_k, \boldsymbol{C}_{\mathbf{y}}^{(1)}(\mathbf{x}_k^{(1)}, \mathbf{u}_k)) \end{cases} . \tag{A.4}$$

The above 2 steps conclude stage 1 of learning, i.e., learning intrinsic behaviorally relevant dynamics $\mathbf{x}_k^{(1)}$. Next we explain the (optional) remaining stage 2, which can learn any remaining dynamics in neural activity $\mathbf{x}_k^2$.

**Stage 2**:

**2a** We learn a second recurrent neural network (*RNN2*) with $n_2 := n_x - n_1$ states, to minimize the MSE loss of predicting the residual neural activity, i.e., $\mathbf{y}_k' := \mathbf{y}_k - \boldsymbol{C}_{\mathbf{y}}^{(1)}(\mathbf{x}_k^{(1)}, \mathbf{u}_k)$, given past neural activity and inputs (equation A.5). States of *RNN2*, i.e., $\mathbf{x}_k^{(2)}$, together with $\mathbf{x}_k^{(1)}$ from stage 1 constitute the full neural dynamics, i.e., $\mathbf{x}_k = \begin{bmatrix} \mathbf{x}_k^{(1)} & \mathbf{x}_k^{(2)} \end{bmatrix}^T$. This optimization step can be formulated as:

$$\begin{cases} \mathbf{x}_{k+1}^{(2)} &= \boldsymbol{A}^{(2)}(\mathbf{x}_k^{(2)}) + \boldsymbol{K}^{(2)}(\mathbf{y}_k, \mathbf{u}_k, \mathbf{x}_k^{(1)}) \\ \mathbf{y}_k' &= \boldsymbol{C}_{\mathbf{y}}^{(2)}(\mathbf{x}_k^{(2)}, \mathbf{u}_k) \\ loss &: MSE(\mathbf{y}_k', \boldsymbol{C}_{\mathbf{y}}^{(2)}(\mathbf{x}_k^{(2)}, \mathbf{u}_k)) \end{cases} . \tag{A.5}$$

**2b** Finally, another readout, $\boldsymbol{C}_{\mathbf{z}}^{(2)}$, can be learned to map $\mathbf{x}_k^{(2)}$ to the residual behavior, i.e., $\mathbf{z}_k' := \mathbf{z}_k - \boldsymbol{C}_{\mathbf{z}}^{(1)}(\mathbf{x}_k^{(1)}, \mathbf{u}_k)$, to minimizing the overall behavioral loss (equation A.6):

$$\begin{cases} \mathbf{z}_k' &= \boldsymbol{C}_{\mathbf{z}}^{(2)}(\mathbf{x}_k^{(2)}, \mathbf{u}_k) \\ loss &: MSE(\mathbf{z}_k', \boldsymbol{C}_{\mathbf{z}}^{(2)}(\mathbf{x}_k^{(2)}, \mathbf{u}_k)) \end{cases} . \tag{A.6}$$

Note that the optimization in stage 2b does not change *RNN2* or $\mathbf{x}_k^{(2)}$ that were learned in stage 2a. So although stage 2b is supervised by behavior, the second set of states $\mathbf{x}_k^{(2)}$ are still learned unsupervised with respect to behavior.

In case a very low state dimension is specified by the user for stage 1 ($n_1$ lower than the ground truth shared dimensionality), *RNN1* would not have enough capacity to learn all behaviorally relevant dynamics. In that case, some behaviorally relevant neural dynamics will be left for *RNN2* in stage 2 to learn. This is why the $\boldsymbol{C}_{\mathbf{z}}^{(2)}$ transformation from $\mathbf{x}_k^{(2)}$ to behavior is included in the model, to allow such behaviorally relevant information in $\mathbf{x}_k^{(2)}$ to be utilized to improve behavior decoding.

### A.1.3 Non-encoded behavior-specific dynamics

To remove the non-encoded behavior-specific dynamics, as a preprocessing step, we fit a high-dimensional ($n_x = 150$ in all real data analyses) unsupervised RNN to extract neural dynamics alone by minimizing neural prediction MSE (equation A.7):

$$\begin{cases} \mathbf{x}_{k+1}^{(0)} & = & \boldsymbol{A}^{(0)}(\mathbf{x}_k^{(0)}) + \boldsymbol{K}^{(0)}(\mathbf{y}_k, \mathbf{u}_k) \\ \mathbf{y}_k & = & \boldsymbol{C}_{\mathbf{y}}^{(0)}(\mathbf{x}_k^{(0)}, \mathbf{u}_k) \\ loss & : & MSE(\mathbf{y}_k, \boldsymbol{C}_{\mathbf{y}}^{(0)}(\mathbf{x}_k^{(0)}, \mathbf{u}_k)) \end{cases} . \tag{A.7}$$

Then a readout $\boldsymbol{C}_{\mathbf{z}}^{(0)}$ is trained to map these neurally relevant states $\mathbf{x}_k^{(0)}$ to behavior:

$$\begin{cases} \mathbf{z}_k & = & \boldsymbol{C}_{\mathbf{z}}^{(0)}(\mathbf{x}_k^{(0)}) \\ loss & : & MSE(\mathbf{z}_k, \boldsymbol{C}_{\mathbf{z}}^{(0)}(\mathbf{x}_k^{(0)})) \end{cases} \tag{A.8}$$

After parameters of the above are learned, we run inference on the training data to obtain the filtered behavior as output of the preprocessing RNN model (first line in equation A.8) and subsequently use it in place of the original behavior in BRAID (in equations A.1, A.3 , A.6). In simulations, we validate that this additional stage can successfully remove any input-driven behavior dynamics not encoded in the neural recordings (figure A.2). We include this preprocessing step in all reported real data analyses with BRAID.

**Stage 3**: As mentioned in 3.3, if desired, BRAID can also learn behavior-specific dynamics as separate dissociated latent states using a post-hoc learning step. This step is performed after BRAID's main learning is done and is meant to be used in conjunction with BRAID's preprocessing stage explained above. In this post-hoc learning step (stage 3), we first infer the behavior using the originally learned BRAID model. We then obtain the residual behavior ($\mathbf{z}_k''$) by subtracting the inferred behavior from the measured behavior. We then learn a third RNN (*RNN3*) that optimizes the prediction of the residual behavior using *only* the external inputs. The following equations summarize this step:

$$\begin{cases} \mathbf{z}_k'' & := & \mathbf{z}_k - [\boldsymbol{C}_{\mathbf{z}}^{(1)}(\mathbf{x}_k^{(1)}, \mathbf{u}_k) - \boldsymbol{C}_{\mathbf{z}}^{(2)}(\mathbf{x}_k^{(2)}, \mathbf{u}_k)] \\ \mathbf{x}_{k+1}^{(3)} & = & \boldsymbol{A}^{(3)}(\mathbf{x}_k^{(3)}) + \boldsymbol{K}^{(3)}(\mathbf{u}_k) \\ \mathbf{z}_k'' & = & \boldsymbol{C}_{\mathbf{z}}^{(3)}(\mathbf{x}_k^{(3)}, \mathbf{u}_k) \\ loss & : & MSE(\mathbf{z}_k'', \boldsymbol{C}_{\mathbf{z}}^{(3)}(\mathbf{x}_k^{(3)}, \mathbf{u}_k)) \end{cases} . \tag{A.9}$$

We summarize BRAID's three stages and their use-case in table A.1.

### A.1.4 Model architecture details and hyperparameters

Throughout the manuscript, to model nonlinearities within any of the transformations i.e., $\boldsymbol{A}(\cdot)$, $\boldsymbol{A}_{fw}(\cdot)$, $\boldsymbol{K}(\cdot)$, $\boldsymbol{K}_{fw}(\cdot)$, $\boldsymbol{C}_{\mathbf{z}}(\cdot)$, and $\boldsymbol{C}_{\mathbf{y}}(\cdot)$, we use a multi-layer perceptron (MLP), also known as a feedforward neural network, with a single hidden layer, 64 units in the hidden layer, and a *ReLU* nonlinearity as activation function. Otherwise, to keep a transformation linear, we replace the MLP with a linear mapping implementing a matrix multiplication, which is a special case of an MLP with

Table A.1: Summary of the three stages of BRAID and their use-case in learning various dynamics. Each stage has two RNNs: a predictor form and a generator form RNN, denoted without and with a $fw$ subscript, respectively.

| Stage | Models | Functionality |
|---|---|---|
| 1 | $RNN1$, $RNN1_{fw}$ | Intrinsic behaviorally relevant neural dynamics |
| 2 | $RNN2$, $RNN2_{fw}$ | Residual neural-specific intrinsic dynamics |
| 3 | $RNN3$, $RNN3_{fw}$ | Residual behavior-specific dynamics not encoded in neural activity |

no hidden layers and a linear activation function. For example, linear BRAID is a BRAID model with all mappings being linear whereas BRAID *Nonlinear $C_{\mathbf{z}}$* has a nonlinear MLP as the behavior decoder (both $C_{\mathbf{z}}^{(1)}$ and $C_{\mathbf{z}}^{(2)}$) while all of its other transformations are linear.

We use an Adam optimizer (Kingma & Ba, 2017) in all BRAID optimizations. We train models up to a maximum number of 2500 epochs to ensure convergence, while employing early stopping to avoid overfitting. We provide example learning curves in figure A.7. Details of the hyperparameters used for BRAID are provided in table A.2.

Table A.2: BRAID hyperparameters used in real data experiments and simulations

| Hyperparameter | Value |
|---|---|
| Number of hidden layers in nonlinear maps | 1 |
| Number of hidden units in nonlinear maps | 64 |
| Nonlinear activation | ReLU |
| Learning rate | 0.001 |
| Batch size | 32 |
| Sequence length | 128 |
| Optimizer | Adam |

We also show that BRAID can locate the correct structure of nonlinearity within all possible combinations in our simulations (table 2). Here, we set each of the following four groups of transformations i.e., $A(\cdot)/A_{fw}(\cdot)$, $K(\cdot)/K_{fw}(\cdot)$, $C_{\mathbf{y}}(\cdot)$, $C_{\mathbf{z}}(\cdot)$, as linear or nonlinear, resulting in a total of $2^4$ cases. To select one final configuration for the nonlinearity, we follow an automatic nonlinearity selection procedure for a given dataset. In this procedure, within the training data, we perform a 2-fold inner cross-validation in which we fit BRAID models with all $2^4$ nonlinearity configurations and then pick the nonlinearity structure with the best cross-validated behavior decoding on the held-out section *of the training data*. Then we retrain a BRAID model with that selected structure on the entire training data to get our final model. Finally, we evaluate that final model on the unseen test data. We refer to this approach as automatic nonlinearity selection (table 2).

In simulations, state dimensions are set to be the same as that of the true model underlying the data. In real data analyses, to investigate the effect of $n_x$, we vary the state dimension in $n_x \in [1, 2, 4, 8, 16, 32, 64]$ and report the results (figure A.5). For BRAID and DPAD models, we always learn the first 16 dimensions via stage 1 i.e., $\mathbf{x}_k^{(1)} \in \mathbb{R}^{n_1}$ with $n_1 = \min(16, n_x)$, and if there is any more capacity left (i.e., if $n_x - n_1 = n_2$ is positive), it is dedicated to the irrelevant states $\mathbf{x}_k^{(2)} \in \mathbb{R}^{n_x - n_1}$ and is learned using stage 2. We pick 16 as the dimensionality of the behaviorally relevant states as in our experiments, because BRAID reached close to its peak behavior decoding at this dimension. We refer to models with state dimensions $n_x = 16$ and 64 as low and high-dimensional regimes, respectively.

### A.1.5 INTRINSIC DYNAMICS AND EIGENVALUES

To evaluate how well the intrinsic behaviorally relevant neural dynamics are learned, in simulations (figures 2, A.1), we assess the eigenvalues of the generative transition $A_{fw}$ which characterize the intrinsic dynamics. Note that the ground truth and learned models in these simulations both have a linear intrinsic state transition $A_{fw}$ and the nonlinearity either lies in the transformation from latent

states to behavior (figure 2) or transformation from external inputs to the latent space (figure A.1). We learn the forward recursion parameters $\boldsymbol{A}_{fw}$ and $\boldsymbol{K}_{fw}$ of equations 1 and 3 as we optimize multi-step-ahead predictions. We take the eigenvalues of the intrinsic transition $\boldsymbol{A}_{fw}$ and compare them to that of the ground truth model. For the ground truth ($\lambda_i$) and identified ($\hat{\lambda}_i$) eigenvalues we first pair them as $\{\lambda_1, \lambda_2, \ldots, \lambda_{n_1}\}$ and $\{\hat{\lambda}_1, \hat{\lambda}_2, \ldots, \hat{\lambda}_{n_1}\}$ such that the sum of squared distances of pairs is minimized. We then calculate the normalized eigenvalue error as:

$$\frac{\sqrt{\sum_{i=1}^{n_1} \|\lambda_i - \hat{\lambda}_i\|^2}}{\sqrt{\sum_{i=1}^{n_1} \|\lambda_i\|^2}}. \tag{A.10}$$

## A.2 BASELINES

First, to assess the impact of the nonlinearities learned by BRAID, we compare it against two fully linear dynamical methods: IPSID (Vahidi et al., 2024) and linear BRAID. Second, to highlight the significance of modeling the effect of measured inputs on the neural-behavioral dynamics, we take an autonomous dynamical model, DPAD (Sani et al., 2024), as another baseline. Another important aspect of BRAID is supervision of the behaviorally relevant dynamics in presence of inputs in its first stage. To assess prioritization of the behaviorally relevant dynamics due to this supervision, we take an equivalent unsupervised baseline termed U-BRAID detailed below. We also compare BRAID against a multi-modal nonlinear method that allows accounting for external inputs termed mmPLRNN (Kramer et al., 2022). Finally, we compare BRAID to TNDM (Hurwitz et al., 2021), a second autonomous method for modeling neural-behavioral dynamics.

### A.2.1 IPSID

IPSID, similar to BRAID, models the effect of measured external inputs on neural-behavioral dynamics but operates under a fully linear framework. It fits the parameters of a linear version of equation A.1 via a projection-based analytical algorithm called subspace identification (Van Overschee & De Moor, 1996).

### A.2.2 LINEAR BRAID

Linear BRAID serves as another linear baseline, retaining the same architecture and learning stages as BRAID but with all transformations replaced by linear mappings. In essence, linear BRAID and IPSID have a similar model that are learned differently. Linear BRAID uses the same numerical optimization used in BRAID. BRAID reduces to linear BRAID by removing all hidden layers within model transformations and setting all activation functions to linear. In simulations, we find that linear BRAID and IPSID perform similarly as expected (figure 2).

### A.2.3 DPAD

Dissociative Prioritized Analysis of Dynamics (DPAD) (Sani et al., 2024), learns a nonlinear model that dissociates and prioritizes dynamics shared between neural activity and behavior, but it importantly does not account for the external inputs. Originally, DPAD also does not allow for multi-step-ahead optimization and thus does not learn a generative form representation of the dynamics, which is in contrast to the learning of $\boldsymbol{A}_{fw}$ in BRAID. We extend DPAD to add optimization of multi-step-ahead predictions into the DPAD framework for a more fair comparison to BRAID in terms of forecasting.

### A.2.4 U-BRAID

U-BRAID is an unsupervised method, which only performs stage 2 of the BRAID learning procedure. As such, U-BRAID learns all neural dynamics irrespective of their relevance to the behavior, but still while considering inputs (equation A.5). U-BRAID does not utilize behavior information in

learning dynamics, and the extracted latent states are later mapped to the behavior data via a downstream decoder (equation A.6 but without $\mathbf{x}_k^{(1)}$). In fact, U-BRAID is special case of BRAID with $n_1 = 0$ and $n_x = n_2$.

### A.2.5 MMPLRNN

Multi-modal piecewise-linear RNN (mmPLRNN) is a method previously introduced for multi-modal dynamical modeling with piecewise-linear RNNs (Kramer et al., 2022). This method allows for modeling external inputs, although this aspect of it has not been investigated in any prior work. Nevertheless, we compare BRAID to an input-driven mmPLRNN to further assess its performance. mmPLRNN builds on a prior work, PLRNN (Durstewitz, 2017), by fusing information from two modalities (e.g., neural activity and behavior). By design, mmPLRNN utilizes both modalities (and input) during inference, which is in contrast to BRAID that only uses neural activity (and input) during inference. Although this provides the benefit of using more data for behavior decoding to mmPLRNN and confounds the comparison with BRAID, we still include the mmPLRNN results. We train mmPLRNN models with nonlinear readouts comparable to BRAID and compare their neural-behavioral forecasting. mmPLRNN is a generative model whose parameters are learned via variational inference. We used the recommended hyperparameters from the original work[1].

### A.2.6 TNDM

Targeted Neural Dynamical Modeling (TNDM) (Hurwitz et al., 2021) is a method based on sequential autoencoders that learns two sets of dynamics: one contributing to both neural and behavioral data, and the other only contributing to neural data. TNDM uses a non-causal, bidirectional RNN as the encoder to infer the initial conditions for its relevant and irrelevant generator/decoder RNNs. The dynamical model is learned via variational inference with both neural and behavioral reconstructions optimized simultaneously with a combined loss. Unlike BRAID, TNDM does not account for modeling the effect of external inputs on neural-behavioral dynamics. Therefore, for comparisons, we also implement an extension of TNDM to allow the inclusion of external inputs. In this version, we provide the external inputs to TNDM model as input by concatenating them with the neural activity as the input to the model. Importantly, we do not add reconstruction of inputs as part of the loss to keep the loss the same as that of the original TNDM and keep the learned model focused on neural-behavioral reconstruction.

We compare BRAID to TNDM (with and without addition of sensory stimuli as external input) in our real data experiments. Unlike BRAID that models the neural observations with a Gaussian distribution, TNDM uses a Poisson observation model for the neural data. Therefore, in our comparisons to TNDM, we analyze non-smoothed spike counts in 50ms bins (for both TNDM and BRAID). We use the default hyperparameters from the original work[2] for TNDM.

### A.2.7 LFADS

Latent Factor Analysis via Dynamical Systems (LFADS) (Pandarinath et al., 2018) is an unsupervised method that combines nonlinear dynamical modeling with sequential autoencoders. Similar to TNDM, LFADS uses a non-causal, bidirectional RNN as the encoder to infer the initial conditions for a generator/decoder RNN. The dynamical model is learned via variational inference for unsupervised neural reconstruction. LFADS has a version that uses additional RNNs called controller networks to infer unmeasured inputs from the neural data and use those inferred inputs to drive its generator RNN. We use this version of LFADS (O'Shea & Pandarinath, 2021)[3], to serve as an additional benchmark in comparison to BRAID's modeling of measured inputs. For a full visualization of the LFADS model including the controller networks see Supplementary Fig. 12 in Pandarinath et al., 2018. Briefly, without a controller network, all the information about the complete trial has to be encoded into the initial state of the LFADS generator RNN, which then autonomously evolves to extract states and factors over the course of the trial. In contrast, a controller network acts as a regular non-autonomous RNN for LFADS and allows its generator RNN to take corrections from the neural data throughout the trial, hence improving its capacity to accommodate non-smooth changes

---

[1]We use the implementation provided in `https://github.com/DurstewitzLab/mmPLRNN`
[2]We use the implementation provided in `https://github.com/HennigLab/tndm`
[3]We use the implementation provided in `https://lfads.github.io/lfads-run-manager/`

in the middle of trials (Pandarinath et al., 2018). The controller network itself does not directly take neural data as input, rather an additional bidirectional RNN called the controller-encoder operates on the neural data of each trial, and the states of this controller-encoder are passed as input to the controller network. As a result, the 'inferred inputs' in LFADS are also non-causally inferred. To compare results with BRAID, we set the number of factors, and the latent states of the generator and the controller-encoder networks to be the same as BRAID. We also pass the same smoothed neural data to LFADS and use the Gaussian neural loss option accordingly. Other hyperparameters were set as those in the original work (Pandarinath et al., 2018).

### A.2.8   CEBRA

CEBRA (Schneider et al., 2023) is a recent method proposed for extracting latent embeddings from neural data in a way that can be guided by behavior. CEBRA uses a 1-dimensional convolutional network to extract embeddings from small windows of neural data and guides the extraction of latents via a contrastive loss. The supervised version of CEBRA (i.e., CEBRA-Behavior) uses a contrastive loss on behavior to learn embeddings that dissociate samples with different behavior data, and are thus behaviorally relevant. Due to its use of a convolutional network to extract embeddings, CEBRA does not learn explicit recursive dynamics and also extracts embedding from a finite window of neural data at a time. CEBRA also does not learn any models to decode behavior or neural data form the extracted embeddings. Thus, as done in the original work (Schneider et al., 2023), we fit k-NN regression[4] models to decode neural-behavioral data from the extracted CEBRA embeddings. For a fair comparison, we also provided the measured sensory inputs time-series of the task to CEBRA by concatenating them with neural activity as the input. We use the default hyperparameters from the original work[5] for CEBRA.

### A.3   NON-HUMAN PRIMATE ELECTROPHYSIOLOGICAL RECORDINGS FROM

We analyzed a publicly available dataset (O'Doherty et al., 2017) in which a macaque (monkey I) performs a motor task. Spiking activity was recorded from primary motor cortex (M1), while the subject controlled a 2D cursor to reach targets that appeared on random locations on a grid within a virtual reality environment. Targets appeared back to back, without any time gaps. We took the subject's 2D fingertip position and velocity as the behavior time-series $\mathbf{z}_k$, and the sensory input, taken as 2D location of the current target, as the input signal $\mathbf{u}_k$. We analyzed the first spike dimension available for each channel-resulting in 89 to 92 units from the first 7 available recording sessions and randomly selected half of these units to model as our neural activity. For neural modality, we use spike counts within 50 ms non-overlapping windows. Finally, we smoothed the spike counts by a Gaussian kernel with a 50 ms s.d. (except for in table A.5) and took that as the neural time-series $\mathbf{y}_k$. We report the mean and standard error of the mean (s.e.m.) computed across 7 sessions and 5 cross-validated folds.

As a second neural modality, we also model the raw local field potential (LFP) activity recorded from the same monkey during the same task. As the only preprocessing, we apply a 10 Hz anti-aliasing filter to be able to downsample the raw LFP to the sampling rate of behavior (i.e., 20 Hz). We refer to this modality as raw LFP data. Results for raw LFP data (figure A.3) were consistent with those obtained for spiking activity (figure 3).

### A.4   SIMULATION DETAILS

We analyze three simulated datasets based on dynamical systems. In all the three, we generate 10 different sets of random linear matrices for equation A.11, then generate the ground truth latent states $\mathbf{x}_k$, neural activity $\mathbf{y}_k$ and behavior observations $\mathbf{z}_k$. In equation A.11, $\mathbf{w}_k$, $\mathbf{v}_k$, and $\boldsymbol{\epsilon}_k$ are zero-mean white Gaussian noises accounting for unmeasured excitations, neural observation noise, and behavioral observation noise respectively. $f_{C_z}(\cdot)$ and $f_B(\cdot)$ are nonlinear functions, as described below for each simulation.

---

[4]sklearn.neighbors.KNeighborsRegressor (Pedregosa et al., 2011)

[5]We use the implementation provided in `https://github.com/AdaptiveMotorControlLab/CEBRA`

$$\begin{cases} \mathbf{x}_{k+1} & = & \boldsymbol{A}_{fw}\mathbf{x}_k + f_B(\boldsymbol{B}\mathbf{u}_k) + \mathbf{w}_k \\ \mathbf{y}_k & = & \boldsymbol{C}_{\mathbf{y}}\mathbf{x}_k + \boldsymbol{D}_{\mathbf{y}}\mathbf{u}_k + \mathbf{v}_k \\ \mathbf{z}_k & = & f_{C_z}(\boldsymbol{C}_{\mathbf{z}}\mathbf{x}_k) + \boldsymbol{D}_{\mathbf{z}}\mathbf{u}_k + \boldsymbol{\epsilon}_k \end{cases} \tag{A.11}$$

To generate a temporally structured input in all cases, we simulate a separate random linear state space model according to equation A.12 and take its output as the external input $\mathbf{u}_k$ to the main model of equation A.11.

$$\begin{cases} \mathbf{x}_{k+1}^{\mathbf{u}} & = & \boldsymbol{A}_{fw}^{\mathbf{u}}\mathbf{x}_k^{\mathbf{u}} + \mathbf{w}_k^{\mathbf{u}} \\ \mathbf{u}_k & = & \boldsymbol{C}_{\mathbf{u}}\mathbf{x}_k^{\mathbf{u}} + \mathbf{v}_k^{\mathbf{u}} \end{cases} \tag{A.12}$$

### A.4.1 SIMULATION 1: SPIRAL BEHAVIOR MANIFOLD

For the first simulation, we generate data with a spiral behavior manifold. To do so, we apply a pointwise nonlinear mapping $f_{C_z}(\nu) = \left[ \frac{\bar{\nu}}{2} \cos(\bar{\nu}) \quad \frac{\bar{\nu}}{2} \sin(\bar{\nu}) \right]^T$ to the readout from the latent states (third line in equation A.11). The bar over the function input $\nu$ indicates a scaling factor that normalizes it before nonlinear function is applied. See figure 2a for a visualization of the nonlinearity. We take $f_B(\cdot)$ as identity, set dimensions to $n_y = n_z = n_u = n_x = n_1 = 2$, and take $D_{\mathbf{y}} = D_{\mathbf{z}} = 0$ for this simulation.

### A.4.2 SIMULATION 2: TRIGONOMETRIC BEHAVIOR MANIFOLD

For the second simulation with trigonometric behavior map, we apply another nonlinearity, pointwise sinusoidal nonlinear function, $f_{C_z}(\nu) = a\sin(\bar{\nu}) + b\bar{\nu}$, to the latent states to generate the behavior $\mathbf{z}_k$. See figure 2e for a visualization of the nonlinearity. In this simulation $f_B(\cdot)$ is taken as identity function and we set $n_y = n_z = n_u = n_x = n_1 = 1$.

### A.4.3 SIMULATION 3: TRIGONOMETRIC INPUT-ENCODER

For the third simulation, as we iterate over the state equation (first line in equation A.11), we apply a pointwise sinusoidal nonlinear function, $f_B(\nu) = a\sin(\bar{\nu}) + b\bar{\nu}$, to the input. See figure A.1a for a visualization of the nonlinearity. Here $f_{C_z}(\cdot)$ is taken to be an identity function. In this simulation, we set $n_y = n_z = n_u = n_x = n_1 = 1$.

We also perform two additional simulations (figure A.2) that are similar in nonlinearity structure to the second and third simulations explained above, but incorporate an additional 1-dimensional latent state $\mathbf{x}_k^{(3)}$, as in figure 1a, representing input-driven behavior-specific dynamics not encoded in the neural activity, as follows:

$$\begin{cases} \begin{bmatrix} \mathbf{x}_{k+1}^{(1)} \\ \mathbf{x}_{k+1}^{(3)} \end{bmatrix} & = & \begin{bmatrix} \boldsymbol{A}_{fw}^{(1)}\mathbf{x}_k^{(1)} \\ \boldsymbol{A}_{fw}^{(3)}\mathbf{x}_k^{(3)} \end{bmatrix} + \begin{bmatrix} f_B(\boldsymbol{B}^{(1)}\mathbf{u}_k) \\ \boldsymbol{B}^{(3)}\mathbf{u}_k \end{bmatrix} + \mathbf{w}_k \\ \mathbf{y}_k & = & \boldsymbol{C}_{\mathbf{y}}^{(1)}\mathbf{x}_k^{(1)} + \boldsymbol{D}_{\mathbf{y}}\mathbf{u}_k + \mathbf{v}_k \\ \mathbf{z}_k & = & f_{C_z}(\boldsymbol{C}_{\mathbf{z}}^{(1)}\mathbf{x}_k^{(1)}) + \boldsymbol{C}_{\mathbf{z}}^{(3)}\mathbf{x}_k^{(3)} + \boldsymbol{D}_{\mathbf{z}}\mathbf{u}_k + \boldsymbol{\epsilon}_k \end{cases}. \tag{A.13}$$

### A.5 SUPPLEMENTARY RESULTS

### A.5.1 BRAID CAN EXCLUDE NON-ENCODED BEHAVIOR-SPECIFIC DYNAMICS

Here, we demonstrate that the optional preprocessing step in BRAID (detailed in section 3.3) can dissociate behavior-specific dynamics (i.e., those that are not encoded in the neural activity) during learning and make sure they are not conflated with intrinsic neural dynamics and are not mixed into the neural states ($\mathbf{x}_k^{(1)}$ and $\mathbf{x}_k^{(2)}$). We conducted two additional simulations similar in structure to the second and third simulations explained in section 4.1.2. However, here, for all simulated models, we added input-driven dynamics that influenced behavior but were not encoded in neural activity (denoted as $\mathbf{x}_k^{(3)}$ in figure 1a and equation A.13). The preprocessing step is intentionally expected to yield latent states that are potentially less predictive of behavior, but are encoded in neural activity. When desired, BRAID provides the option to further learn behavior-specific dynamics post-hoc

with a separate latent state $(\mathbf{x}_k^{(3)})$. The preprocessing and post-hoc learning steps allow BRAID to avoid conflation of non-encoded behavior dynamics with others, while also being able to learn these dynamics and thus not incurring any overall reduction in behavior decoding.

We fitted BRAID models with the preprocessing, and both with and without post-hoc learning of behavior-specific dynamics. With the preprocessing, BRAID reached the neural prediction performance of the ground truth model indicating correct removal of behavior-specific dynamics (figure A.2). Moreover, the optional learning of behavior-specific dynamics led to reaching the behavior decoding performance of the ground truth model (figure A.2), suggesting that one could optionally learn these dynamics as well within BRAID to gain interpretability (by learning a disentangled model) without compromising decoding performance.

### A.5.2 Additional Supplementary Tables

In this section we include additional supplementary tables that further support the results from the main text. The caption for each supplementary table includes all the details, but here we provide a list of these tables:

- Table A.3: Ablation analysis showing the importance of the $RNN_{fw}$ generative model in BRAID for learning intrinsic dynamics.
- Table A.4: BRAID results for different nonlinearity configurations in real NHP data.
- Table A.5: Comparison with TNDM in real NHP data.
- Table A.6: Comparison with LFADS and CEBRA in real NHP data.
- Table A.7: BRAID results for modeling different number of neurons in real NHP data.
- Table A.8: Same as table 3 shown in terms of the $R^2$ metric.

Table A.3: **A separate generative model is essential to learn the intrinsic dynamics accurately with BRAID**.
Row 1: BRAID, when learning a separate generative model ($RNN_{fw}$), more accurately learns the intrinsic dynamics as quantified by the error in identifying eigenvalues of the ground truth intrinsic dynamics in the simulated dataset with spiral manifold in figure 2a.
Row 2: The error when ablating the forward RNN from BRAID.
Rows 3-4: DPAD, even when optimized with $m$-step-ahead prediction loss and/or with input ($\mathbf{u}_k$), does not learn the intrinsic dynamics accurately.

| Method | $\log_{10}$ normalized eigenvalue error |
|---|---|
| BRAID | **-1.3963 $\pm$ 0.2551** |
| BRAID without $RNN_{fw}$ | 0.0635 $\pm$ 0.2212 |
| DPAD + $m$-step loss | 0.0357 $\pm$ 0.1587 |
| DPAD + input ($\mathbf{u}_k$) + $m$-step loss | -0.6512 $\pm$ 0.0354 |

Table A.4: Comparison of BRAID model's nonlinearity configurations in the NHP dataset ($n_x = n_1 = 16$, 4-step-ahead).

| Model nonlinearity | Behavior forecasting CC | Neural forecasting CC |
|---|---|---|
| Linear | 0.7453 $\pm$ 0.0066 | 0.1767 $\pm$ 0.0054 |
| Recursion ($\boldsymbol{A}, \boldsymbol{A}_{fw}$) | 0.7121 $\pm$ 0.0059 | 0.2719 $\pm$ 0.0061 |
| Encoder ($\boldsymbol{K}, \boldsymbol{K}_{fw}$) | 0.7181 $\pm$ 0.0078 | 0.1646 $\pm$ 0.0049 |
| Decoder ($\boldsymbol{C_z}, \boldsymbol{C_y}$) | **0.8042 $\pm$ 0.0085** | **0.3274 $\pm$ 0.0078** |

Table A.5: Comparison to TNDM, both when sensory input is additionally provided to the TNDM model and when it is not (see appendix A.2.6). All models are learned in low-dimensional regime, i.e., BRAID with $n_x = n_1 = 16$, and TNDM with 16 relevant factors only. We used non-smoothed spike counts as the neural signals in this analysis. BRAID performances are for causal 1-step-ahead prediction, whereas the TNDM performances are non-causal smoothing performances, which are the only option for TNDM since it is a sequential autoencoder.

| Method | Behavior decoding CC | Neural prediction CC |
|---|---|---|
| TNDM | $0.3752 \pm 0.0170$ | $0.3021 \pm 0.0051$ |
| TNDM with sensory input | $0.6219 \pm 0.0103$ | $\mathbf{0.3075 \pm 0.0050}$ |
| BRAID (ours) | $\mathbf{0.7841 \pm 0.0079}$ | $0.2935 \pm 0.0053$ |

Table A.6: Comparison to LFADS (with controller), and CEBRA (CEBRA-behavior). LFADS inferred the external input to the dynamical system (appendix A.2.7). For CEBRA, w.s.i. (with sensory input) indicates that sensory input is additionally provided to the model to obtain the embeddings (appendix A.2.6). All models are learned with both low-dimensional ($n_x = 16$) and high-dimensional ($n_x = 64$) latent states ($n_1 = 16$ for BRAID). BRAID performances are for causal 1-step-ahead prediction, whereas LFADS and CEBRA performances are non-causal smoothing, and 0-step-ahead reconstruction respectively.

| Method | Behavior decoding CC | | Neural prediction CC | |
|---|---|---|---|---|
| | $n_x = 16$ | $n_x = 64$ | $n_x = 16$ | $n_x = 64$ |
| LFADS | $0.4714 \pm 0.0192$ | $0.5891 \pm 0.0135$ | $\mathbf{0.5615 \pm 0.0086}$ | $0.5920 \pm 0.0072$ |
| CEBRA w.s.i. | $0.7544 \pm 0.0073$ | $0.7514 \pm 0.0072$ | $0.5070 \pm 0.0053$ | $0.5693 \pm 0.0041$ |
| BRAID (ours) | $\mathbf{0.8109 \pm 0.0074}$ | $\mathbf{0.8085 \pm 0.0076}$ | $\mathbf{0.5571 \pm 0.0051}$ | $\mathbf{0.8401 \pm 0.0061}$ |

Table A.7: **Effect of the number of neurons on BRAID's performance.** Results of the BRAID modeling when different numbers of neurons included as the neural signal. We included neurons from the same channel sets across different sessions and the ranges indicate the number of neurons available from those channels across different sessions. In all 3 rows, neural predictions are evaluated on the same common set neurons (i.e., the smallest set shown in row 1) to make the neural forecasting results comparable across rows. Behavior forecasting improved with more neurons and neural forecasting remained largely stable. These results suggest that BRAID can aggregate behaviorally relevant information across larger populations of neurons, while still being able to model this higher-dimensional population activity well.

| Scale | Behavior forecasting CC | | Neural forecasting CC | |
|---|---|---|---|---|
| | $n_x = 16$ | $n_x = 64$ | $n_x = 16$ | $n_x = 64$ |
| 20-21 neurons | $0.7727 \pm 0.0091$ | $0.7709 \pm 0.0089$ | $0.3206 \pm 0.0081$ | $0.4115 \pm 0.0094$ |
| 41-43 neurons | $0.8042 \pm 0.0085$ | $0.7970 \pm 0.0086$ | $0.3220 \pm 0.0088$ | $0.4202 \pm 0.0091$ |
| 89-92 neurons | $0.8337 \pm 0.0061$ | $0.8302 \pm 0.0072$ | $0.3329 \pm 0.0082$ | $0.4195 \pm 0.0088$ |

Table A.8: $R^2$ results for the same analyses provided in table 3. Forecasting performance (4-step-ahead) compared to baselines in NHP dataset for models with low-dimensional ($n_x = 16$) and high-dimensional ($n_x = 64$) latent states. $n_1 = 16$ for BRAID, linear BRAID, and DPAD. 1 of the 7 sessions were excluded from this table due to a very negative outlier in the mmPLRNN results.

| Method | Behavior forecasting $R^2$ | | Neural forecasting $R^2$ | |
|---|---|---|---|---|
| | $n_x = 16$ | $n_x = 64$ | $n_x = 16$ | $n_x = 64$ |
| linear BRAID | $0.5821 \pm 0.0056$ | $0.5763 \pm 0.0084$ | $0.0239 \pm 0.0036$ | $0.1519 \pm 0.0067$ |
| DPAD | $0.4561 \pm 0.0141$ | $0.5497 \pm 0.0101$ | $0.0321 \pm 0.0042$ | $0.1344 \pm 0.0069$ |
| U-BRAID | $0.6047 \pm 0.0097$ | $\mathbf{0.6684 \pm 0.0088}$ | $\mathbf{0.1768 \pm 0.0068}$ | $\mathbf{0.1860 \pm 0.0064}$ |
| mmPLRNN | $0.4737 \pm 0.0118$ | $0.6127 \pm 0.0369$ | $0.0734 \pm 0.0079$ | $0.0563 \pm 0.0670$ |
| BRAID (ours) | $\mathbf{0.6680 \pm 0.0118}$ | $\mathbf{0.6578 \pm 0.0129}$ | $0.1083 \pm 0.0060$ | $\mathbf{0.1792 \pm 0.0064}$ |

### A.5.3 ADDITIONAL SUPPLEMENTARY FIGURES

In this section we include supplementary figures that further support the results from the main text. The caption for each supplementary figure include all the details, but here we provide a list of these supplementary figures:

- Figure A.1: Simulation results for simulation 3 with a trigonometric nonlinear input-encoder.
- Figure A.2: Validation of the optional third stage of learning in BRAID for learning behavior-specific input driven dynamics.
- Figure A.3: Results of modeling the raw LFP modality in the real NHP data.
- Figure A.4: Average low-dimensional latent state trajectory extracted from real NHP data.
- Figure A.5: Results in real NHP data for different latent state dimensions.
- Figure A.6: Example BRAID decoded time series in real NHP data.
- Figure A.7: Example plot showing loss versus epochs for BRAID.

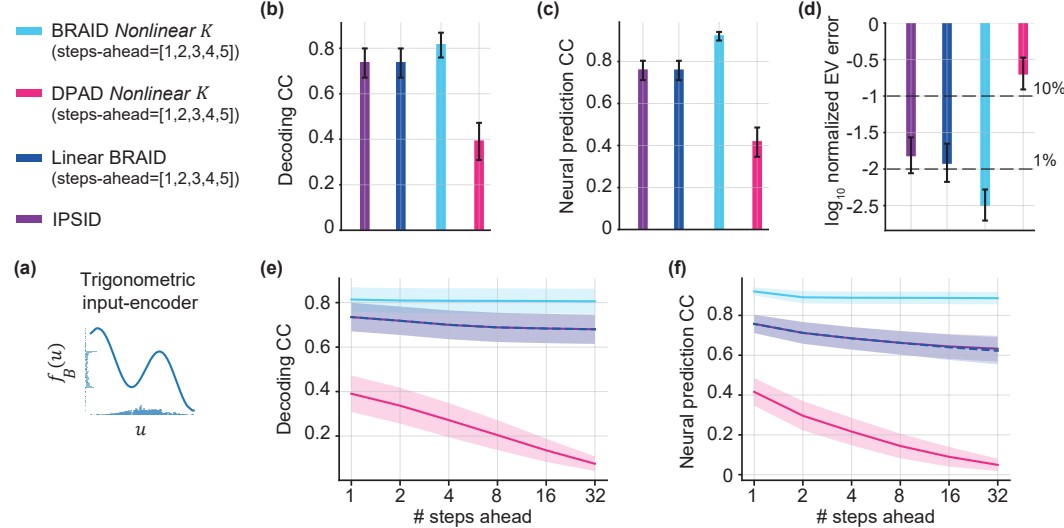

Figure A.1: **BRAID results, optimized for forecasting, in simulation with trigonometric input-encoder.** **(a)** Visualization of example nonlinearity in the simulation. **(b-c)** 1-step-ahead behavior decoding and neural prediction for nonlinear BRAID, nonlinear DPAD, linear BRAID, and IPSID. **(d)** Error in identifying intrinsic dynamics of the true model, quantified by the eigenvalues of the state transition matrix $A_{fw}$. **(e-f)** Behavior and neural forecasting accuracy for 1 to 32 steps ahead, enabled by learning the intrinsic dynamics ($A_{fw}$), with predictions optimized for $[1, 2, 3, 4, 5]$-steps-ahead (section 3.1).

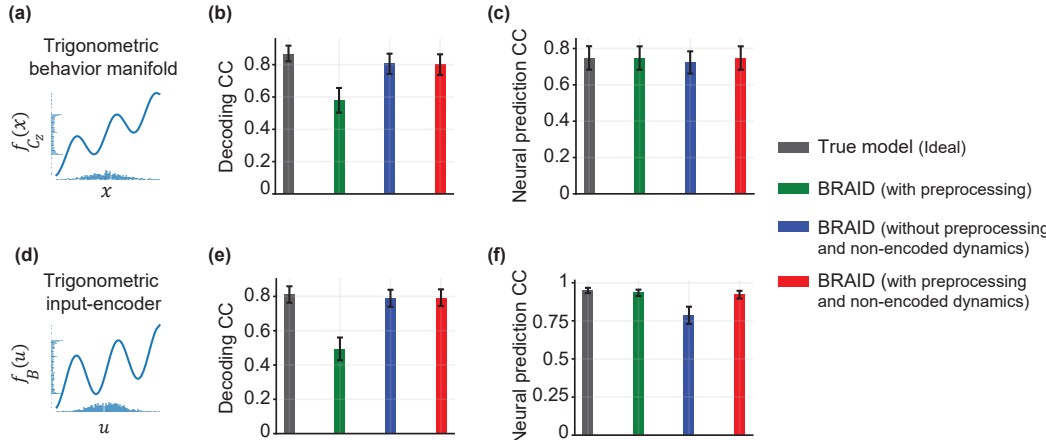

Figure A.2: **Behavior preprocessing successfully excludes non-encoded, behavior-specific dynamics in simulations**. 1-step-ahead behavior decoding and neural predictions for simulations with **(a-c)** trigonometric behavior decoder, and **(d-f)** input-encoder. Note that the true model includes behavior-specific dynamics, so here we expect that decoding of BRAID with preprocessing but without the post-hoc learning of behavior-specific dynamics (shown as green), to be worse than that of true model. Once the post-hoc learning step is also performed (shown as red), BRAID reaches ideal performance, but importantly does so while these behavior-specific dynamics are dissociated into a separate latent state $C_{\mathbf{z}}^{(3)}$. In contrast, without the preprocessing step (shown as blue), BRAID reaches ideal decoding performance, but does so without having dissociated behavior specific dynamics to not be included in $C_{\mathbf{z}}^{(1)}$. See section A.1.3 for details.

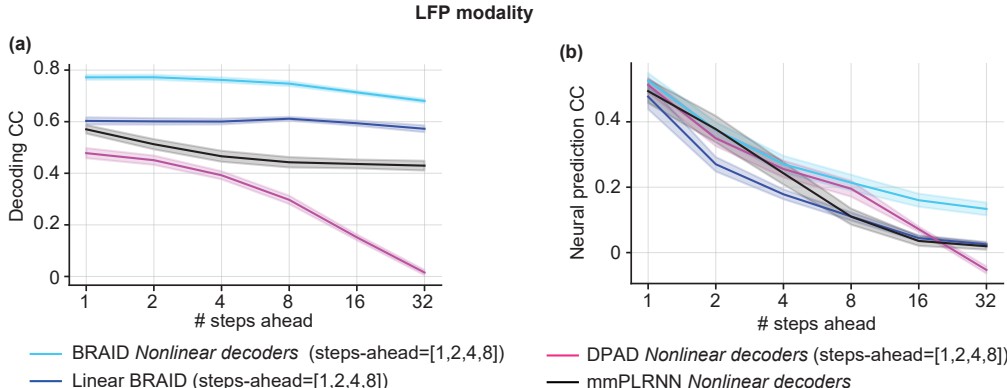

Figure A.3: **Analysis of Local Field Potential (LFP) neural modality. BRAID outperforms baselines in neural-behavioral forecasting**. We used LFP neural data as a second different modality and performed analysis similar to the one in figure 3 for smoothed spike counts. **(a)** Behavior and, **(b)** neural activity forecasting correlation coefficient (CC) for BRAID, linear BRAID, DPAD, and mmPLRNN for $n_x$=16. Shaded areas show the s.e.m., across the 3 recording sessions and 5 cross-validation folds. NHP results in all other figures and tables are for spiking data.

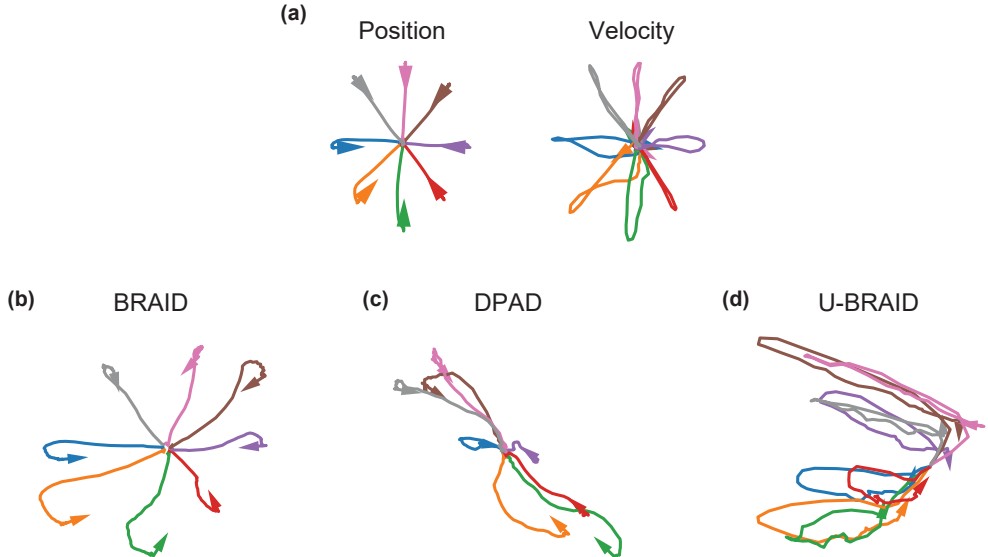

Figure A.4: **Latent states trajectories revealed on non-human primate reaching dataset**. We divide reach trials to 8 conditions based on reach direction (shown by colors) and find the condition-averaged (4-step-ahead) latent states trajectories for 2 dimensional models. **(a)** Condition-averaged behavior i.e., movement position and velocity. **(b-d)** Condition-averaged latent state trajectories for (b) BRAID, (c) DPAD and (d) U-BRAID. BRAID learns the most well-separated posterior (latent) trajectories for the 8 conditions in the task. U-BRAID's trajectories are the least separated, showing that BRAID is more successful in extracting the behaviorally relevant intrinsic dynamics that are more congruent with behavior. Also, DPAD's trajectories are not as well-separated as BRAID's, although they are more separated than the unsupervised version (U-BRAID) as expected.

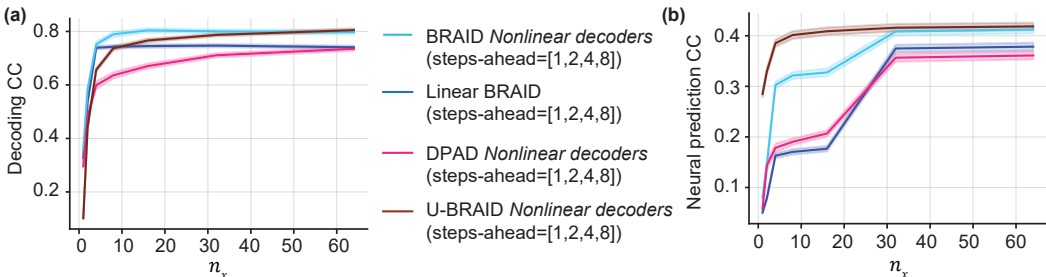

Figure A.5: **Forecasting (4-step-ahead predictions) correlation coefficient across latent dimensions.** BRAID predicts both **(a)** behavior and **(b)** neural activity more accurately than DPAD due to modeling input, and than linear BRAID due to modeling nonlinearity. As state dimension increases beyond 16 (dedicated to behaviorally relevant dynamics, i.e., $n_1 = 16$), BRAID uses stage 2 to learn neural specific dynamics, thus reaching the unsupervised baseline, i.e., U-BRAID, in neural prediction. For BRAID and DPAD, the first 16 state dimensions are dedicated to the behaviorally relevant neural dynamics (i.e., $n_1 = 16$) while any remaining dimensions ($n_x > 16$) are dedicated to the residual non-shared neural dynamics. 4-steps-ahead in this dataset corresponds to 200ms ahead.

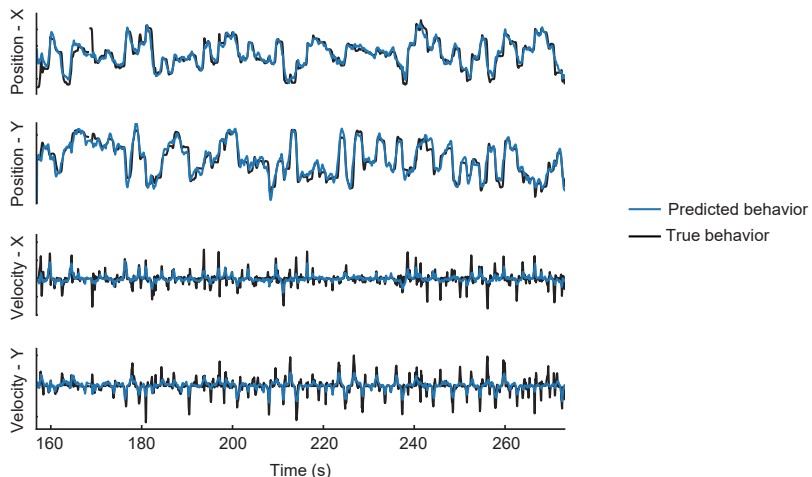

Figure A.6: **Predicted behavior visualization**. True behavior versus BRAID's (4-step-ahead) predicted behavior for a representative session corresponding to the results in table 3 ($n_x = 64$).

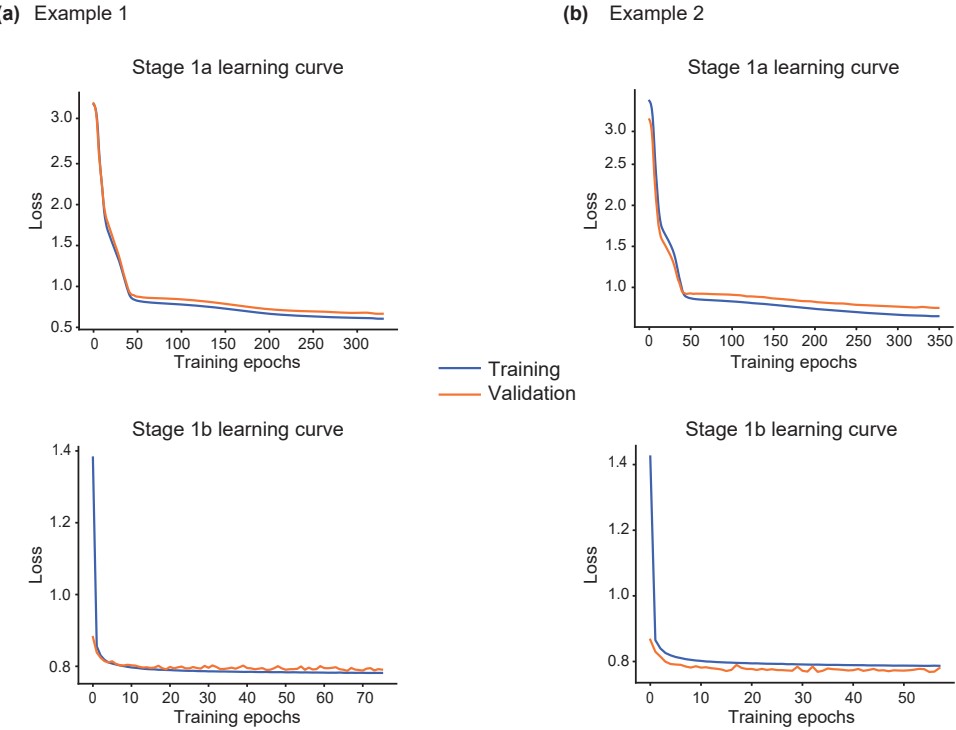

Figure A.7: **Learning curve examples for BRAID model**. Loss values as a function of number of epochs for training and validation datasets for two example runs of BRAID. Top row: learning curve examples for the *RNN1*, *RNN1$_{fw}$*, i.e., stage 1a, in section 3.2. Bottom row: learning curves examples for stage 1b in section 3.2. These examples are taken from the same models whose performances are reported in figure 3 and table 3.

