# OpenReview forum: "BRAID: Input-driven Nonlinear Dynamical Modeling of Neural-Behavioral Data"
_ICLR.cc/2025/Conference — ICLR 2025 Poster_

### Official Review · Reviewer_kFWw · 2024-10-31

**Soundness:** 3
**Presentation:** 3
**Contribution:** 3
**Rating:** 6
**Confidence:** 4

**Summary:**

This paper introduced a novel framework, BRAID, modeling nonlinear neural and behavioral dynamics with external input. This model dissociates behaviorally relevant neural dynamics, neural specific and behavioral specific dynamics, and outperforms baselines.

**Strengths:**

BRAID considers input-driven neural and behavioral dynamics and dissociates them. The authors performed ablation studies to shown which parts of BRAID contributed to the neural and behavior forecasting.

**Weaknesses:**

1. Although the model has been shown to be highly efficient in modeling neural spiking data, it has not been tested on other modalities, such as widefield calcium imaging.
2. Although the authors mentioned that their model is not to infer unmeasured input, I still think this may be a weakness of this model, because (for example) the neural and behavioral dynamics can be encoded by unmeasured input.

**Questions:**

1.	In Figure 1, should there be a connection between stage 2 and behavior Z_k if you are learning C_z(2) in Stage 2 2b? If so, what is the difference between X_k(1) and X_k(2) since both are connected to neural activity via [C_y(1), C_y(2)] and to behavior via [C_z(1), C_z(2)]?
2.	In equation 1, the model can learn behavior dynamics that are predictable from the input but are not encoded in the recorded neural activity, but how do you make sure that the behavior prediction is not dominated by the input? In Figure 3a, there is a high correlation between your input target position and your behavior cursor position and velocity. If so, can the model learn any information encoded in neural activity?
3.	The model also has an input term in modeling neural activity y, what does this input mean? Because you have an input in latent space that encodes intrinsic contribution, then you also have the same input to encode input-driven contribution.
4.	How do you define your initial states x0?
5.	I am curious, how do you ensure that the model learned all encoded information and the residual is the non-encoded information since the model is nonlinear with high flexibility?
6.	Following Q5, you mentioned you trained the model until convergence, could you show your learning curve against epochs with zooming in the last epochs if necessary?

---

> ### Author Response · Authors · 2024-11-20
> **[W1,2]: Applying method to other modalities and unmeasured inputs**
>
> **[W1] Other modalities:**
>
> We thank the reviewer for their suggestion, and we agree that applying our method to other neural modalities would be valuable. Here we focused on analyzing spiking activity as it is one of the most widely used modalities in neuroscience and neurotechnology. In addition to spiking activity, we have extensive synthetic data using which we validated BRAID. Nevertheless, we applied BRAID to another neural modality, that is local field potential (LFP) (see  Fig. B3 in the temporary supplementary file), and found results consistent with those obtained on spiking activity. We will include these new results in the paper.
>
> **[W2] Unmeasured inputs:**
>
> We appreciate the reviewer’s great insight on this point, and acknowledge the value of the complementary approach of input inference in lines 508-513. We should clarify that, while input inference is not the goal of BRAID, similar to other latent models, BRAID also has the capacity to implicitly learn unknown inputs as part of the latent states, as long as these unmeasured inputs encode neural and behavioral dynamics as the reviewer notes. This is because BRAID’s training objective is neural and behavioral prediction. As such, BRAID can automatically dedicate part of its latent states to learning unknown inputs if that benefits neural-behavioral prediction. In that respect, BRAID is not fundamentally less capable than input-inference approaches. The main difference is that in BRAID we do not attempt to explicitly disentangle unmeasured inputs or put any constraints such as smoothness on them.
>
> Adding potential assumptions on the nature of unmeasured inputs as done in input-inference approaches can be an interesting future direction for BRAID to further disentangle unmeasured inputs. We will clarify this by revising our discussion.

---

> ### Author Response · Authors · 2024-11-20
> **Questions - Part 1**
>
> **[Q1] C_z(2) and the difference between X_k(1) and X_k(2)**:
>
> Thank you for this observation. The reviewer raises a good point, which we clarify here. In Fig. 1a, our aim is to visualize the objectives and supervisions for learning dynamics/states rather than the complete computation graph. While we do learn a mapping $C_z^{(2)}$ from $x^{(2)}$ to $z$, this mapping is learned in a final optimization step, meaning after x1 and x2 are fully learned and fixed through stages 1 and 2. We clarify this point in lines 810-813 and explain why/when it could be beneficial in lines 814-818. $x^{(1)}$ and $x^{(2)}$ are different even if each has a mapping to both behavior ($C_z^{(1)}$, $C_z^{(2)}$) and neural activity ($C_y^{(1)}$, $C_y^{(2)}$). This is because during training, $x^{(1)}$ is learned to optimize the training objective of behavior forecasting in stage 1. Only after $x^{(1)}$ is learned, in stage 2 of training, $x^{(2)}$ is learned to optimize a distinct training objective, which is the forecast of residual neural activity (i.e., neural activity that is not already predicted by $x^{(1)}$). As such, $x^{(1)}$ learns the intrinsic behaviorally relevant neural dynamics and $x^{(2)}$ learns the other/residual intrinsic neural dynamics.
>
>
> **[Q2]: How do you make sure that the behavior prediction is not dominated by the input? Can the model learn any information encoded in neural activity?**
>
> We thank the reviewer for this great comment. Addressing the problem raised by the reviewer is precisely the motivation behind the preprocessing stage introduced by BRAID explained under section 3.3. Note that we refer to such non-encoded dynamics as behavior-specific dynamics as well (Fig. 1a, sections 3.3 and 4.1.3). Briefly, the preprocessing stage avoids such non-encoded behavior dynamics by learning an RNN to filter out any behavior dynamics that are predictable by the input but are not encoded in the neural activity. That way, the output of the preprocessing RNN is a preprocessed behavior signal and both stages 1 and 2 of training use this preprocessed behavior rather than the original behavior. Thus, this preprocessing stage restricts the dynamics learned in the forthcoming stages 1, 2 to be encoded in neural activity (see section 3.3). In section 4.1.3, we validate this step in numerical simulations showing that when the preprocessing is included, the learned model has an improved neural prediction matching the one obtained via the true model (Fig A.2). This means that BRAID did indeed learn information encoded in neural activity. Importantly also, in all real data analyses we incorporate this preprocessing RNN stage. We clarify this in line 272 and line 841. Moreover, in real data results, we report neural prediction performance in addition to behavior decoding to highlight the fact that our method not only predicts behavior well, but also outperforms baselines without sensory input in terms of neural prediction. This better neural prediction shows that the model is able to learn dynamics encoded in neural activity rather than being dominated by the non-encoded/behavior-specific input dynamics.
>
>
> **[Q3]:  input term in modeling neural activity y, what does this input mean?**
>
> The input terms in the decoders ($C_z$/$C_y$) are implemented and included in the formulation for the sake of generality (to support the applications where feedthrough is of interest) but we have not used them in real data analyses as we are interested in the predictability through the latent states. In general, the direct input term to observations (feedthrough term) can sometimes be helpful for discrete-time systems (Åström 2013). Since the dynamic equation takes the input up to the previous time-step to infer the current state and the current observation, in general, a feedthrough term can be helpful for providing concurrent information from the input during the length of the current time-step, which has not already been aggregated into the current latent (is not in the inputs from previous time-steps).
>
>
> Reference:
>
> Åström, Karl J., and Björn Wittenmark. 2013. Computer-Controlled Systems: Theory and Design, Third Edition. Courier Corporation.
>
>
> **[Q4]: How do you define your initial states x0?**
>
> We use zero initialization i.e., set x0 to zero. This is because we process the data in relatively long sequences, which makes the results minimally dependent on the initial state since its effects typically die off within a few samples. We will clarify this in the paper.

---

> ### Author Response · Authors · 2024-11-20
> **Questions - Part 2**
>
> **[Q5]: how do you ensure that the model learned all encoded information and the residual is the non-encoded information since the model is nonlinear with high flexibility**
>
> Our objective function during training is optimizing for learning encoded information. Specifically, our training objectives are prediction of neural activity and prediction of encoded-behavior (behavior after the preprocessing step, see response to question 2). So if our objective is optimized and converges, then it implies that we have learned most of the encoded information. As with all deep learning methods, we use standard numerical optimization with stochastic gradient descent for learning. We continue the gradient descent up to 2500 epochs or until the loss stops improving for several epochs (i.e., early stopping mechanism). In our results, the early stopping criteria is typically met at a much lower number of epochs suggesting that convergence is relatively fast.
>
> Additionally, during behavior preprocessing, we ensure all encoded behavior information is learned by fitting a high-dimensional preprocessing model (RNN) to neural data which is then used to maintain only the encoded behavioral information while filtering out non-encoded information.
>
>
> **[Q6]: Could you show your learning curve against epochs with zooming in the last epochs if necessary**
>
> We have added example learning curves (see Fig. B4 in the temporary supplementary file) to show convergence of our models. We will add these results to the paper.

---

> > ### Comment · Reviewer_kFWw · 2024-11-22
> >
> > I would like to thank the authors for their responses, and I appreciate all the new results. I retain my positive score of 6, as I think the new proposed method does perform better than baseline models. The reason that I did not increase my score is the result is increased but not significantly, so we may not be able to get new insights from this model. In summary, I still hold a positive but borderline score. Once again, I thank the authors for addressing my concerns.

---

> > > ### Author Response · Authors · 2024-11-25
> > >
> > > We sincerely thank the reviewer for their positive evaluation of our work and truly appreciate their feedback. We also wanted to provide two updates and clarify one point.
> > >
> > > First, we have now updated the manuscript by incorporating the temporary supplementary file with all the new analyses into it, so the updated manuscript now addresses all comments from the reviewer.
> > >
> > > Second, in terms of performance improvements, we have now also compared to two additional state-of-the-art baselines, i.e., LFADS (Pandarinath et al 2018) and CEBRA (Schneider et al 2023), showing that BRAID outperforms them in neural-behavioral prediction (please see table A.6, also copied in response to reviewer FaeG).
> > >
> > > Third, we would like to emphasize that our primary goal is  to address the critical challenge of disentangling intrinsic neural-behavioral dynamics and input dynamics by modeling external inputs rather than simply improve performance. We do so by developing a multi-stage learning framework, with each stage dissociating one of three subtypes of neural-behavioral dynamics: i) intrinsic behaviorally relevant neural dynamics, ii) other intrinsic neural dynamics, iii) behavior-specific dynamics. Further, we show that to disentangle intrinsic dynamics, each stage needs a predictor and a generator model, with the latter learning the intrinsic dynamics with a forecasting loss (table A.3). Therefore, our work not only enhances performance but more importantly also provides a framework for dissociating subtypes of intrinsic neural-behavioral dynamics and disentangling them from input dynamics, which is critical for basic neuroscientific investigations.

---

### Official Review · Reviewer_XENV · 2024-11-03

**Soundness:** 4
**Presentation:** 3
**Contribution:** 3
**Rating:** 8
**Confidence:** 4

**Summary:**

The authors tackle the problem of disentangling intrinsic from input-driven neural dynamics underlying behavioral measurements. They do so through a novel nonlinear model called BRAID. Their approach is to conceptualize the intrinsic as the generative (i.e. forward) dynamics, and the input-driven as the (posterior-) predictor dynamics. The authors model each of these components with nonlinear DNNs for flexibility, allowing each to be learned. Importantly, they devise a multi-stage training procedure that prioritizes learning behavior-relevant dynamics, with neural reconstruction placed second.  They show how this approach can help disentangle the neural dynamics directly relevant to behavior, in both synthetic experiments and monkey motor reaching neural activity.

**Strengths:**

- The work tackles a very relevant question in the field of statistical neuroscience of disentangling the relationships between behavior, neural activity and inputs, and shows originality in doing so with careful modeling.
- The modeling is well-formulated in the main text, and made clear for the reader in the appendix and through provided code.
- The decomposition into multiple stages and sub-components to encourage learning behavior-relevant activity is interesting and novel (to my knowledge).
- The authors showcase their model on synthetic and real experimental data, showing strong performance in both.
- The authors help support the significance of their results by (1) comparing them against many baselines and extensive ablations of their model, and (2) performing many runs, providing error bars for all numerical results.
- The metrics are meaningful and bypass common non-identifiability problems (such as considering the eigenvalues of A)
- The figures are clean and easy to understand.

**Weaknesses:**

I am putting a score of 6 (marginal accept) but I would be willing to increase it if the authors can help address my weaknesses/questions, and in particular the "major" ones below.

EDIT: The authors have sufficiently addressed the weaknesses below -- I have updated my score accordingly.

Major:
- I am not convinced by the claims of real-time inference in the discussion, and the lack of surrounding literature on the predictor components of the model. Variational inference approaches for sequential models similarly model the (sequential, i.e. filtering) posterior and optimize it jointly with the generative model in the ELBO. I believe the similarities could be actually quite exact, in which case a fairer depiction of the relationship with VI is necessary. On that note, Table 1 refers to your method as pure LL optimization, but perhaps it could be more easily interpreted through an ELBO objective.
- Most of the results are numerical, and the paper lacks a bit in alternate results such as posterior trajectories or visual reconstruction.
- In line with the previous comment, the monkey experiment results are numerical and would benefit from some post-training analysis if the goal is to show the model as a modeling and analysis tool.

Medium:
- Appendix should be for additional but not necessary details to understand the paper. One example of this is the simulations in section 4.1. The notation from L312 follows the appendix where you introduce $f_{C_z}$ and $\nu, \bar\nu$ -- I would make it consistent with eq. (1) or add eq (A.11) to the main text.
- Identifiability: the authors discuss behavioral- vs neural-relevant activity, but having nonlinear A, Cs are another type of inter-dependence, which could be discussed further.
- Monkey experiment: numerical performance results are good but sometimes lack transparency in their presentation. For instance, U-BRAID does do better on neural, but that is by construction and does not put BRAID in any lesser light. The corresponding paragraph (3 of section 4.2) however does not acknowledge this better performance. Similarly, the authors refer to mmPLRNN as having an "unfair advantage" (L451). I would remove this.
- The relationship with probabilistic formulations is skimmed but still alluded to with the ELBO/LL in Table 1. I would expand further on these ideas.

Minor:
- Using $\cdot$ (\cdot) instead of $.$ (dot) is more standard for place-holder variables in functions (L191, L312, L1015)
- Dependency on x^1 on Stage 2 could be made more explicit in the text
- Table 2: bold entries per metric would be more representative

**Questions:**

Some of the following questions relate directly to the weaknesses raised above.
- How does your predictor relate to sequential posterior in variational inference?
- Monkey experiment: What do x^1 and x^2 look like in this task? Is there any non-encoded activity x^3?
- Could you provide more details on the "automatic" selection (L313)? Is it simply picking the best performing?

---

> ### Author Response · Authors · 2024-11-20
> **Major [W1] Real-time inference and distinctions from variational inference and ELBO**
>
> We thank the reviewer for their positive feedback on our work and appreciate their openness to raise their score. We now address their weaknesses/questions below.
>
>
> The reviewer raises an interesting point regarding the analogy between BRAID’s predictor-generative structure and the encoder-decoder architecture of the VAEs.  Below are several key distinctions:
>
> Causality and real-time inference:
>
> First, we apologize for the use of the word “real-time”, which is unclear. By real-time, we mean that inference is done causally in time; in particular, inferring the state at time $k$ only uses neural observations up to time $k$ but none of the future neural observations. Furthermore, this inference is also recursive and thus computationally efficient. Recursive means that BRAID can infer the state at the next time step based on the inferred state in the present time-step (without redoing any computation on past neural observations and by just updating the present inferred state based on the present neural observation). BRAID can perform inference causally in time and recursively because its predictor and generative models are both recursive **unidirectional** RNNs and generate predictions given the past neural observations. In other words, BRAID’s RNNs all run only forward in time. This is in contrast with SOTA sequential VAEs such as TNDM and LFADS, which require neural observations for an entire trial/segment to approximate the posterior distribution at the initial point and subsequently any time-step in the trial/segment using a **bidirectional** RNN (i.e., thus their inference is done non-causally in time). The causal (unidirectional) and recursive features of inference in BRAID can be of utmost interest for real-time decoding applications such as brain-computer interfaces (BCIs). We illustrate the causality of BRAID through its computation graph in Fig. 1b. We will change the word real-time to explain that we mean causal (unidirectional) and recursive.
>
> Objective:
>
> In general, a fundamental distinction of BRAID with variational methods is that the latter have *the statistical parameters* of a latent distribution (e.g., mean and variance of a Gaussian) as the ultimate output of the encoder, from which the decoder samples a point to initiate decoding. As such, variational methods also often include a KL-divergence term in the loss to enforce the distribution of this variational latent variable. In contrast, BRAID does not have a KL-divergence term, nor does it have such a sampling mechanism between the encoder and decoder, and is thus not a variational approach. BRAID learns all parameters (predictor and generative) through direct optimization of m-step ahead prediction (i.e., forecasting), which is quantified by the m-step-ahead LL. We will note both the analogy and the distinction in objective in the manuscript.

---

> ### Author Response · Authors · 2024-11-20
> **Major [W2,3]: Most of the results are numerical, and the paper lacks a bit in alternate results such as posterior trajectories or visual reconstruction**
>
> We thank the reviewer for this important suggestion. We now include two categories of alternative results (see Figs. B1,2  in the temporary supplementary file): 1) predicted behavior trajectories i.e., position and velocity of the arm in the non-human primate dataset,  2) predicted low-dimensional posterior trajectories (i.e., latent state trajectories) for visualization. We can see that BRAID is accurate in predicting behavior. BRAID also has the most well-separated latent trajectories for different reach directions in the task. U-BRAID’s trajectories are the least separated, showing that BRAID is more successful in extracting the behaviorally-relevant dynamics that are more congruent with behavior. Also, DPAD’s trajectories are also not as well-separated as BRAID’s, though they are more separated than the unsupervised version as expected.

---

> ### Author Response · Authors · 2024-11-20
> **Medium [W3, W4]: Additional details in the main text and identifiability**
>
> [W3]: We thank the reviewer for this note and will revise the manuscript to make sure all important definitions are provided in the main text.
>
> [W4]: We agree with the reviewer that identifiability is a general challenge for nonlinear latent state models including BRAID and beyond. In practice, different nonlinearities can be useful and easier to learn depending on the specific dataset. For example, while making all model elements nonlinear may also be a solution, making only some parameters nonlinear may be equally accurate, but more parsimonious and thus may result in a better fit for a given finite number of training samples. While our results suggest that in our dataset having nonlinear decoders provides superior performance compared to other nonlinearities, this might not be the case in another dataset. In real data, therefore, one can search for optimal nonlinearity based on the desired performance. Indeed, BRAID enables this search through an automatic selection so that users won’t need to manually check different options for nonlinearity. We show that the source of nonlinearity in the ground-truth simulated data can be identified by the automatic nonlinearity selection in BRAID, which selects the best performing configuration (section 4.1.1, line 314, and Table 2). We will add this point to the discussion.

---

> > ### Author Response · Authors · 2024-11-20
> > **Medium [W5, W6]: Transparency in presentation and wording and Relationship with probabilistic formulation and ELBO vs LL**
> >
> > [W5]: We thank the reviewer for bringing this point to our attention. Following the reviewer’s feedback, we will explicitly add to the text that, as expected, U-BRAID outperforms BRAID in terms of neural prediction when used in a low-dimensional regime as, by construction, it focuses on optimal neural prediction instead of behavior decoding. We will also remove the “unfair advantage” phrasing for mmPLRNN. What we meant here was that mmPLRNN uses not only the neural activity and sensory inputs, but also the behavior in reconstructing behavior while BRAID just uses neural activity and sensory inputs; so mmPLRNN having access to more data is a confound of the comparison, despite which BRAID outperforms mmPLRNN.
> >
> >
> > [W6]: We thank the reviewer for the insight regarding analogy with variational inference and will add a discussion to the manuscript to explain the differences between VI and BRAID’s objective (please refer to the response to major [W1]).

---

> ### Author Response · Authors · 2024-11-20
> **Minor weaknesses**
>
> **Notation error**:
>
> We thank the reviewer for their correction and will fix the place-holder notation throughout the paper.
>
>
> **Dependency on x^1 on Stage 2 could be made more explicit in the text**:
>
> We thank the reviewer for this remark. We now add a clarification on this in the text/caption. We also show this in Fig. 1a by an arrow from x1 to x2.
>
> **Bold entries per metric would be more representative**:
>
> Our emphasis was on highlighting cases with neural-behavior prediction being simultaneously accurate, to make it clear that other rows do either behavior or neural prediction well, but not both at the same time. We will add a notation (underline) to also highlight models that perform well for an individual metric.

---

> ### Author Response · Authors · 2024-11-20
> **Questions**
>
> **[Q1]: How does your predictor relate to sequential posterior in variational inference?**
>
> Please refer to the response to major [W1].  VI learns the posterior by optimizing ELBO loss, which is a combination of a KL divergence w.r.t. a fixed prior and LL term. BRAID learns the posterior, i.e., the latent states of the predictor model, only based on past observations (m-step or more in the past) by directly optimizing the m-step LL.
>
>
> **[Q2]: Monkey experiment: What do x^1 and x^2 look like in this task? Is there any non-encoded activity x^3?**
>
> We have added a new figure as explained under major [W2,3] to visualize the latent trajectories for the monkey experiment (see Fig. B1 in the temporary supplementary file). BRAID  trajectories (Fig. B.1b) correspond to the behaviorally relevant states i.e., $x^{(1)}$ while trajectories identified by the unsupervised U-BRAID (Fig. B1d) correspond to the irrelevant states i.e., $x^{(2)}$. We do not learn any behavior-specific states ($x^{(3)}$) in any of our real data analyses since our primary goal is to learn dynamics encoded in neural activity. As expected, $x^{(1)}$ are much more well-separated for different behavior conditions compared to $x^{(2)}$ which shows successful learning of the behaviorally relevant dynamics by BRAID.
>
> **[Q3]: Could you provide more details on the "automatic" selection (L313)? Is it simply picking the best performing?**
>
> We thank the reviewer for this question. Yes, this automatic selection procedure selects the best-performing model with an inner-cross validation within the training set. Here is how it works: We explore all possible nonlinearity configurations resulting in $2^4$ cases (each parameter can be either linear or nonlinear). We split the training data into two sections, train models for each case on the first section of training data and evaluate them in the held-out section of training data. We then select the configuration yielding the best behavior prediction on the held-out section of training data. Among options with the same behavior prediction, we select the one with the best neural prediction. Once the best performing nonlinearity is selected, we retrain a model with that configuration using the entire training data and report its performance on the unseen test data. Currently, we describe the automatic nonlinearity selection procedure in lines 314-316 and provide further detail in lines 884-890. We will expand this section to clarify this process as described above.

---

> > ### Comment · Reviewer_XENV · 2024-11-22
> >
> > Thank you to the authors for their replies and clarifications. I am happy to see their comments on our questions and the additional experiments provided in the supplementary file -- the manuscript will stand stronger for their integration. Particularly, I find Figure B.1.(b) of the BRAID posterior compelling.  Now, in response to the Major [W1] comment, I believe using "causal" makes the exposition in the paper clearer, I encourage the authors to use it! However, on that front, my original question remains.
> >
> > First regarding the posterior, as I mentioned, my objection lies in the claims of novelty in the use of sequential filtering posteriors. While I appreciate that most approaches and SOTA methods indeed rely on **bidirectional** RNNs to parameterize the posterior, **filtering** posteriors with **unidirectional** RNNs are not new. We can refer to this review on Dynamical VAEs (https://arxiv.org/abs/2008.12595), where they broadly study the problem of variational inference in sequence modeling. Some models mentioned of direct interest are the Deep Kalman Filters from (Krishnan et al, 2015), and importantly, the Stochastic RNN (Fraccaro et al. (2016)) in Section 9. The latter uses a state-space model, along with an additional RNN that uses the observations as inputs. This seems to be your graphical model, except for the missing dependency on inputs. This comes from a search on the more "deterministic" RNNs literature, and more can be found in the probabilistic variational filtering literature (e.g. Luk et al, ICML 2024). This connects to the next point.
> >
> > Furthermore, while VAEs indeed model the parameters of the latent posterior distribution and sample from it, this is generally more an implementation device (e.g. the reparametrization trick). Variational inference refers to the broad approach of parametrizing the posterior for unobserved variables and optimizing it along with the training. The specific KL term mentioned comes from using the ELBO as the objective for variational inference.
> >
> > In short, I think the claims of (1) filtering posterior novelty and (2) no relationship to variational inference are wrong. This does not discredit the model or approach generally, which I still believe is original, but I maintain that these aspects should be addressed.

---

> > > ### Author Response · Authors · 2024-11-23
> > >
> > > We thank the reviewer for their response and are glad that the reviewer finds our new results, especially the visualization of the latent trajectories compelling and believe that the manuscript is made stronger with their addition. We also thank the reviewer for these excellent references and the insightful discussion about VI. We should clarify that we do not mean to make either of the two claims that the reviewer disagrees with and will make this explicit in the text.
> > >
> > >
> > > First, regarding claim 1 of causality, we apologize if the writing or our responses came across as claiming posterior filtering as a novelty – we agree that many models (e.g., RNNs) have posterior filtering and do not intend to make such a claim. All we meant to say was that unlike several of SOTA models in *neuroscience* like LFADS, TNDM etc, our inference is causally done. So we are just noting a difference compared with these specific models rather than claiming novelty. We will change that sentence to the below to clarify:
> > >
> > > >“Finally, similar to many prior works (Fraccaro et al 2016, Chung et al 2015, Krishnan et al, 2015, Luk et al 2024), BRAID has a causal formulation and can perform inference by recursively inferring the next sample of the latent state after each new observation sample is measured. This is distinct from some nonlinear approaches in neuroscience (Gao et al., 2016; Pandarinath et al., 2018; Hernandez et al., 2020; Hurwitz et al., 2021; Keshtkaran et al., 2022; Gondur et al., 2024; Karniol-Tambour et al., 2024) that perform inference non-causally in time. As such BRAID may also be a good candidate for real-time decoding applications such as brain-machine-interfaces.”
> > >
> > > Regarding claim 2, we do not claim that there is no relationship to variational inference and will update the manuscript to discuss this point. Specifically, we agree that our predictor-generative network even in the case of a vanilla single-stage of our method is analogous to the encoder-decoder networks in VI’s in that we generate our forecasts based on the latent states estimated by the predictor. However, we wanted to clarify that  we do not parametrize the posterior for our unobserved variables or impose a distribution on them, which as the reviewer noted is done in variational inference. We will add the following discussion to clarify this point and refer to it from table 1 to avoid confusion:
> > >
> > >
> > >
> > > >“To disentangle input dynamics from intrinsic dynamics and dissociate the subtypes of neural-behavioral dynamics, BRAID consists of three stages, with each stage having a predictor and a generator model. In each stage, BRAID’s predictor model in that stage infers the latent states, which are subsequently employed as input to compute m-step-ahead predictions via the associated generative model. BRAID’s predictor and generative models are learned jointly to maximize the m-step-ahead log-likelihood. This has analogies to the encoder-decoder architectures such as those commonly used in variational inference (Fraccaro et al 2016, Luk et al 2024, Chung et al 2015, Krishnan et al 2015). The predictor RNN in BRAID has a role similar to that of the encoder in variational inference and the generator RNN has a role similar to that of the decoder in variational inference. However, in variational inference, the posterior distribution of the unobserved variables given the data is parameterized and a part of the optimization loss aims to enforce that distribution on the inferred latent variables (Kingma et al 2013). In contrast, we do not impose such parameterization on the latent states in BRAID. Developing variational methods with the same multi-section architecture and multi-stage learning as BRAID is an interesting future direction to explore.”
> > >
> > > Finally, we appreciate that the reviewer agrees that our model is original. Indeed, the novel contribution of BRAID is enabling the disentanglement of 3 subtypes of dynamics – intrinsic behaviorally relevant neural dynamics, other intrinsic neural dynamics, and behavior-specific dynamics – through a multi-stage learning procedure. Developing variations of BRAID with objectives that are closer to VI can be an interesting future direction. We really appreciate the reviewer’s insightful discussion on this point.

---

> > > > ### Comment · Reviewer_XENV · 2024-11-26
> > > >
> > > > Thank you to the authors for their engagement in this discussion. I have read the updated manuscript and I am satisfied with the updates and corrections. I have increased my score.

---

### Official Review · Reviewer_FaeG · 2024-11-04

**Soundness:** 3
**Presentation:** 2
**Contribution:** 3
**Rating:** 8
**Confidence:** 3

**Summary:**

This paper proposes a new deep learning method to jointly model neural and behavioural data by explicitly modelling task inputs, in order to better model and disentangle input effects and intrinsic neural dynamics that are predictive of behaviour. The learning of the intrinsic dynamics is enabled by the use of a forecasting objective, i.e., $m$-step-ahead neural and behavioural prediction. The method involves 2 (or 3) RNN models, and optimisation is done in multiple stages: a pre-processing stage to filter out behaviours that are not relevant to recorded neural dynamics (used instead of actual behaviour for training), a stage to learn the neural dynamics predictive of behaviour, and a stage to learn any residual neural dynamics. The proposed method outperforms several baselines at predicting neural activity and decoding behaviour, and is also amenable to real-time predictions due to its causal formulation.

**Strengths:**

* The method is well-motivated given prior work, and the causal formulation makes it amenable to real-time inference, which is a strong asset.
* The figures are neat and the experiments are comprehensive. The writing is mostly clear, but I have some comments on explaining the method in a slightly clearer manner (see Weaknesses).
* The method performs well not only on synthetic tasks designed to show its efficacy over several baselines, but also on decoding real neural data.

**Weaknesses:**

* The model architecture and the learning stages are quite complicated, and the writing makes this hard to understand in some places. If I've understood this correctly, there are mainly two RNNs: one to predict the next timestep neural activity given current activity and observed inputs, and another to predict activity $m$ timesteps into the future using just observed inputs. Each of these RNNs' parameters are split into up to 3 subsets, which are optimised sequentially: one to learn behaviourally relevant neural dynamics, one to learn any residual dynamics, and another to learn behaviour that is not encoded in the neural dynamics. This is in addition to an RNN that preprocesses inputs.

  While this preprocessing RNN has been ablated for, can the authors comment on how one identifies in practice whether or not to use the second and third stages of training (RNN2 and RNN3), which were mentioned to be optional? In general, some additional clarity in the writing here would be appreciated.

* It would be important to see how the method scales with the number of neurons – based on the details in the appendix, it seems like the maximum dimensionality explored here (in the neural data case) is around 45. Perhaps the authors could run an experiment comparing decoding performance and neural predicitivity for differing numbers of neurons from the same dataset to show this (and also comment on the time taken to train BRAID).

* From the experiments it seems that BRAID is mainly a single-session model. It is well-known that there can be a lot of variability in neural activity across sessions as animals learn to perform the task better or due to some representational drift. This does not seem to be addressed in the paper as far as I can see, but could the authors comment on how BRAID generalises to unseen sessions, and also specifically for the later parts of a session when training on the initial parts (one of the 5 folds)?

* While the experiments are comprehensive and baselines have been compared against, I think comparisons with LFADS (and if possible, CEBRA) could be useful here – both these approaches seem slightly less involved in terms of training complexity, but would represent baselines when ablating for explicit input modelling (LFADS) or explicit dynamics modelling (CEBRA). The idea here is also that LFADS is designed to infer inputs to the dynamical system, so it might perform better than the extended TNDM baseline included currently.

**Questions:**

* Could the authors attempt to make the writing clearer with regards to explaining the model and the various training stages? Perhaps a summary as a table might help.
* Is it possible to show results when ablating for the second and third (post-hoc) stages of training? Apologies if I've missed this. Also, could the authors comment on what happens when just training the model end-to-end?
* Could the authors consider experiments that show the scaling performance of the model vs number of neurons, and attempt to compare with LFADS (and CEBRA) if possible?
* Could the authors comment on multi-session models, generalisation to unseen sessions?
* Apart from neural predictivity and behaviour decoding, could the authors perhaps use a similarity metric to explicitly compare the learned and true dynamical systems (e.g. using [Dynamical Similarity Analysis](https://openreview.net/forum?id=7blSUMwe7R))? This should be fairly easy for the synthetic tasks, and should be possible in case of the neural data as well.
* It would be interesting to see the model's performance on a dataset involving multiple brain regions, but I understand that this would take more time to run and depend on the availability of a dataset.
* I spotted a potential typo on Line 073 ("prepossessing" -> "preprocessing"?), and another minor issue on Line 111 ("intrinsic representation of dynamic." -> "dynamics"?).

---

> ### Author Response · Authors · 2024-11-20
> **[W1] How one identifies in practice whether or not to use the second and third stages of training and additional clarity on writing of method**
>
> We thank the reviewer for bringing up the clarity issue for use cases of RNN2 and RNN3. The reviewer’s understanding of our method architecture (2 RNNs, 3 subsets of states in each, preprocessing, etc.) is exactly correct, which we truly appreciate. Our primary focus is to learn intrinsic behaviorally relevant neural dynamics with RNN1 and with priority. We will revise the manuscript to carefully explain the use cases for the optional neural-specific (RNN2) and behavior-specific (RNN3) components, which we describe below in detail:
>
> *RNN2*: One would use this RNN if, beyond intrinsic behaviorally relevant neural dynamics, they are also interested in learning the other neural dynamics, which are not shared with behavior (i.e., residual neural dynamics). Since all neural dynamics may not be relevant to the measured behavior, learning RNN2 helps explain the neural-specific dynamics not already captured in RNN1, which learns the shared neural-behavioral dynamics. Note that BRAID allows the two subtypes of neural dynamics (behaviorally relevant vs. other) to be dissociated, as one subtype is learned in RNN1 and one is learned in RNN2. So a user can actually study these subtypes separately. We have demonstrated this use case in a high-dimensional regime where we fit both RNN1 and RNN2 to the real datasets, leading to improved neural predictions due to learning both subtypes of neural dynamics (Table 3, nx=64).
>
> *RNN3*: One would use this RNN3 if they are interested in studying the dynamics in behavior that are not encoded in the recorded neural activity (i.e., the behavior-specific dynamics). RNN3 disentangles behavior-specific dynamics from the two subtypes of intrinsic neural dynamics learned in RNN1 and RNN2. Furthermore, one would use RNN3 if they are interested in improved behavior decoding by using behavior-specific latents. We show this in simulated datasets (Fig. A2). Since our main interest lies in identifying neural dynamics, we do not employ this RNN3 in our real data analysis and always decode behavior purely from neural activity alone. RNN3 is meant to complement the preprocessing stage where the behavior-specific dynamics are filtered out via a separate preprocessing RNN so that RNN1 and RNN2 can exclude those dynamics. We also explain this use-case in section 3.3, lines 273-278 as well as section 4.1.3, lines 375-377.

---

> ### Author Response · Authors · 2024-11-20
> **[W2] Method scaling with number of neurons**
>
> We thank the reviewer. We will perform an analysis on this and report back.

---

> ### Author Response · Authors · 2024-11-20
> **[W3] Generalizability across sessions and for the latter part of the session when trained on the initial parts**
>
> We appreciate this thoughtful comment from the reviewer. With regard to performance on the latter section of a session when trained on the initial parts, this is effectively what happens in the last fold of our 5-fold cross validation, where we train the model on the first 80% of the session and evaluate it on the last 20% of the session. To answer the reviewer’s question, we can compare this final fold with fold 3, where the first and last 40% of the session are used for training while the middle 20% is used for evaluation. As expected, the last fold has slightly worse performance than the middle fold, which can be explained by non-stationarities throughout one session:
>
>
> | **Method**       | **Behavior Forecasting CC** \(n_x = 16\) | **Behavior Forecasting CC** \(n_x = 64\) | **Neural Forecasting CC** \(n_x = 16\) | **Neural Forecasting CC** \(n_x = 64\) |
> |-------------------|------------------------------------------|------------------------------------------|----------------------------------------|----------------------------------------|
> | Last fold    |            0.7720±0.0152              |             0.7611±0.0197          |             0.2995±0.0199           |        0.4078±0.0195
> | Middle fold             |       0.8038±0.0187               |            0.8042±0.0194            |        0.3446±0.0188     |     0.4182±0.0212        |
>
>
> Regarding cross-session generalizability, BRAID as presented in our work is designed for the single-session setting. We will add a discussion on how similar approaches taken elsewhere in the literature for other methods could be taken to add cross-session decoding support to BRAID. One key approach is to add session-specific encoding and decoding matrices between the neural data and the input and outputs of the model in each session, while keeping all other model parameters the same across sessions. This will allow data from multiple sessions to be used for training the model and when a new session is given, one can apply the learned model to it by simply using very minimal data to learn just the simple session-specific encoding/decoding matrices for that new session. Beyond cross-session generalizability of the same task, BRAID may also better generalize across different variations of tasks with different sensory instructions (e.g., center-out targets vs. random targets in a grid); as BRAID can account for these differences, it can help with aligning the intrinsic dynamics across tasks. Extending BRAID by these techniques to account for variability across sessions/datasets is indeed an interesting future direction. We will add a discussion in the manuscript.

---

> ### Author Response · Authors · 2024-11-20
> **[W4] Additional comparisons to LFADS (and CEBRA)**
>
> We thank the reviewer for this comment.. We would like to first note a few points. First, we currently do ablation for explicit input modeling by comparing with DPAD, which does not take into account inputs and is actually closer to our architecture than LFADS. Second, DPAD was compared with LFADS in the DPAD paper, suggesting that it improves behavior decoding compared to LFADS. Thus, we expect BRAID to do better than LFADS as it does better than DPAD. Third, regarding CEBRA, the DPAD paper also does comparisons of DPAD with CEBRA, showing that the dynamical modeling in DPAD leads to better neural-behavioral prediction. As BRAID improves upon DPAD, we expect it to also improve upon CEBRA. Nevertheless, we will do our best to run the comparisons with at least one of CEBRA and/or LFADS by the end of the discussion period and report back.

---

> ### Author Response · Authors · 2024-11-20
> **Questions - Part1**
>
> **[Q1]: Could the authors attempt to make the writing clearer with regards to explaining the model and the various training stages? Perhaps a summary as a table might help**:
>
> We thank the reviewer for this excellent suggestion. We will create a summary table, showing how the 2 representations of dynamics are learned by two RNNs (RNN and RNN_fw) and how the states in each RNN are divided into 3 subsets that are learned by a sequential multi-stage optimization. This reviewer’s explanation of these RNNs and stages was quite accurate and succinct, which we will use for inspiration to make the writing more clear as explained under W1.
>
> **[Q2]: Results for ablating second and third stages, and comment on end-to-end training**:
>
> We apologize for the unclarity. Our main results ablate for second and third stages of training and only use stage 1 alone (Fig. 3 and Table 3 with nx=16). We indicate this by setting n1 (dimension of stage 1) equal to nx (full dimension), which indicates that only stage 1 is used in building the model. We refer to this stage-one-only setting as a low-dimensional regime in our real data analyses and explain this in section 4.2 lines 431. We additionally add stage 2 for the high-dimensional regime with nx=64 in Table 3 (that is n1=16 and n2=48). We explain what these settings mean in section 4.2, lines 431-432 but will make this much more clear: “In addition to these results in the low-dimensional …”. As Table 3 shows, adding stage 2 improves neural forecasting while leaving behavior forecasting essentially unchanged by adding the neural-specific latents. The third (post-hoc) stage is not used in real data analyses for the reason explained in response to Weakness 1. We have however used and validated the third stage in our simulated analyses provided in Fig A.2 and denoted that in the legend as “with non-encoded dynamics”. As this figure shows, ablating for stage 3 (denoted “BRAID (with preprocessing)”, green) results in lower behavior decoding compared with having stage 3 (denoted “BRAID (with preprocessing and non-encoded dynamics)”, red) because stage 3 learns behavior-specific dynamics.
>
> End-to-end training without stages: As the reviewer notes, one can train all components simultaneously instead of the multi-stage approach proposed by BRAID (as done by TNDM and mmPLRNN baselines). We want to highlight that the multi-stage training is an important aspect for prioritization of the intrinsic behaviorally relevant neural dynamics in learning. Having separate stages allows for the intrinsic behaviorally relevant dynamics to be learned first by optimizing a behavioral loss, so that these dynamics are not mixed with other neural dynamics, which are learned in stage 2 while fixing RNN1’s parameters. On the other hand, in an end-to-end training, one needs to simultaneously optimize a combined neural-behavioral prediction loss and as such may learn latent states that mix up behaviorally relevant and other neural dynamics, thus not prioritizing the former and not learning the former as accurately. Indeed TNDM and mmPLRNN both use end-to-end training with neural and behavioral terms, and so the superior behavior decoding of BRAID compared to TNDM and mmPLRNN suggests the benefit of multi-stage learning.
>
> **[Q3]: Scaling performance experiments and comparisons to LFADS (and CEBRA) if possible**:
>
> We thank the reviewer for this comment. As with any model, we believe BRAID will improve forecasting with more neurons. To show this, we are now working on running BRAID with different numbers of neurons and will report the results by the end of the discussion period. Please see response to [W4] above regarding CEBRA and LFADS.
>
> **[Q4]: Comment on multi-session models and generalization**:
>
> Please refer to the response to W3.
>
> **[Q5]: Use of similarity metric to compared learned and true dynamics**:
>
> We thank the reviewer for this question. We agree this is an important step for validations and have done so in our simulated datasets by comparing the learned intrinsic dynamical system to the ground truth system. The results are shown in Fig. 2c,g and Fig. A1d. As intrinsic dynamics in a system can be quantified by the eigenvalues of the state transition matrix, we quantify the similarity between dynamics via the accuracy of learning intrinsic eigenvalues. As the results suggest, the eigenvalues of the intrinsic dynamical system are learned much more accurately using BRAID compared to DPAD and linear baseline (IPSID/linear BRAID). As the reviewer suggests this kind of similarity analysis is possible in the synthetic data where the ground truth is known, but as far as we know is not feasible in real data due to the lack of ground truth. As such, neural/behavior decoding are the measures used in real data, which reflect how accurately the underlying neural-behavioral dynamics were learned.

---

> ### Author Response · Authors · 2024-11-20
> **Questions - Part 2**
>
> **[Q6]: Modeling multiple brain regions**:
>
> We appreciate this comment and agree with the reviewer that studying multiple brain regions and their shared vs private dynamics is an interesting topic in computational neuroscience. Indeed BRAID can facilitate such multi-region investigations because it can model the activity of an upstream brain region as a measured input into the downstream region. Exploring the use of BRAID for this application would be a great future direction.
>
>
> **[Q7]: Typographical errors**:
>
> We thank the reviewer for the corrections and will fix them in the manuscript.

---

> > ### Comment · Reviewer_FaeG · 2024-11-20
> >
> > I would like to thank the authors for their response and appreciate their efforts in addressing my concerns. I am mostly satisfied and would be open to increasing my score, pending the results associated with W2.

---

> ### Author Response · Authors · 2024-11-21
>
> > I would like to thank the authors for their response and appreciate their efforts in addressing my concerns. I am mostly satisfied and would be open to increasing my score, pending the results associated with W2.
>
> We are glad the reviewer is satisfied with our revisions and really appreciate their openness to raise their score pending our results on [W2]. We have now completed this analysis and provide the results in the table below. We will also add this to the appendix. As the table shows, behavior forecasting improves with more neurons and neural forecasting remains largely stable. This suggests that our method can aggregate behaviorally relevant information across a larger population of neurons, while still being able to model this higher-dimensional population activity well.
>
>
> | **Scale**       | **Behavior Forecasting CC** \($n_x$ = 16\) | **Behavior Forecasting CC** \($n_x$ = 64\) | **Neural Forecasting CC** \($n_x$ = 16\) | **Neural Forecasting CC** \($n_x$ = 64\) | **Training time (s)** \($n_x$ = 16\) |  **Training time (s)** \($n_x$ = 64\) |
> |-------------------|------------------------------------------|------------------------------------------|----------------------------------------|----------------------------------------|----------------------------------------|----------------------------------------|
> | 20-21 neurons    |       0.7727 ± 0.0091    |          0.7709 ± 0.0089         |            0.3206 ± 0.0081           |      0.4115 ± 0.0094      |    834.70 ± 45.60 (~14 mins)      |    1092.67 ± 55.38 (~18 mins)      |
> | 41-43 neurons     |       0.8042 ± 0.0085    |          0.7970 ± 0.0086         |             0.3220 ± 0.0088           |      0.4202 ± 0.0091      |    976.95 ± 56.75 (~16 mins)      |     1325.63 ± 73.79 (~22 mins)      |
> | 89-92 neurons     |       0.8337 ± 0.0061    |          0.8302 ± 0.0072         |            0.3329 ± 0.0082           |       0.4195 ± 0.0088      |    934.26 ± 40.33 (~16 mins)      |     1223.02 ± 50.58 (~20 mins)      |

---

> > ### Comment · Reviewer_FaeG · 2024-11-22
> >
> > Thank you for the update. In light of the authors' rebuttal I am inclined to increase my score. Once again I thank the authors and appreciate their efforts in addressing the reviewers' concerns, and I look forward to seeing the updated manuscript.

---

> ### Author Response · Authors · 2024-11-25
>
> >Thank you for the update. In light of the authors' rebuttal I am inclined to increase my score. Once again I thank the authors and appreciate their efforts in addressing the reviewers' concerns, and I look forward to seeing the updated manuscript.
>
> We are grateful that the reviewer is inclined to increase their score in light of our rebuttal and new analyses. We have now updated the manuscript by incorporating the temporary supplementary file with all the new analyses into it. The updated manuscript now includes all suggestions from the reviewer, including the new results for the effect of the number of neurons on our method's performance that the reviewer had asked for. Furthermore, we have now completed the comparisons to two additional baselines suggested by the reviewer, i.e., LFADS (with controller which infers input) and CEBRA (supervised by behavior and with sensory inputs (w.s.i.)). We have now also added these results to the uploaded revised manuscript and copy them below for convenience.
>
> | Scale       | Behavior decoding CC \($n_x$ = 16\) | Behavior decoding CC \($n_x$ = 64\) | Neural prediction CC \($n_x$ = 16\) | Neural prediction CC \($n_x$ = 64\) |
> |-------------------|------------------------------------------|------------------------------------------|----------------------------------------|----------------------------------------|
> | LFADS    |      0.4714 ± 0.0192    |          0.5891 ± 0.0135         |            **0.5615 ± 0.0086**           |      0.5920 ± 00.0072      |
> | CEBRA w.s.i.    |      0.7544 ± 0.0073    |          0.7514 ± 0.0072         |             0.5070 ± 0.0053           |      0.5693 ± 0.0041     |
> | BRAID (ours)     |       **0.8109 ± 0.0074**    |          **0.8085 ± 0.0076**         |            **0.5571 ± 0.0051**           |       **0.8401 ± 0.0061**     |
>
> As the results suggest, BRAID outperformed both new baselines in neural-behavioral prediction, except for low-dimensional models where the neural prediction is on par with LFADS. Note that this is expected as LFADS is an unsupervised model that purely optimizes neural reconstruction while low-dimensional BRAID optimizes behavior prediction to prioritize the intrinsic behaviorally relevant dynamics.
>
> We again sincerely thank the reviewer for their valuable insights and discussions, which we believe significantly improved our manuscript. We are also grateful that the reviewer is inclined to raise their score.

---

> > ### Comment · Reviewer_FaeG · 2024-11-26
> >
> > Thank you for the update, the additional comparisons are appreciated. I maintain my positive opinion of the work and have increased my score.

---

### Official Review · Reviewer_QzSK · 2024-11-09

**Soundness:** 2
**Presentation:** 3
**Contribution:** 2
**Rating:** 3
**Confidence:** 5

**Summary:**

BRAID aims to distinguish the effect of measured inputs towards neural dynamics, while dissociating shared neural-behavioral dynamics, neural-only dynamics, and behavior-only dynamics. It does so by training a series of recurrent neural networks: (1) Stage 1 trains a shared recurrent neural network that outputs both neural activity and behavior for (a) 1-step ahead and (b) multi-step ahead prediction, (2) Stage 2 does similarly but for neural dynamics only, and (3) Stage 3 does so for behavioral dynamics only. The authors show better correlation coefficients with the neural and behavioral activity in validation datasets as compared to some baseline comparisons.

**Strengths:**

- The method dissociates neural-only, vs. behavior-only, vs. shared neural-behavioral spaces.
- BRAID has the ability to take in measured inputs. However, this is not providing any conceptual advance from existing methods, since they are simply fed into the RNNs with an additional input transformation in the form of K.

**Weaknesses:**

- BRAID is very similar to DPAD (Sani et al., 2024), but with the addition of measured inputs, behavior-only dynamics, and an additional network that predicts m-steps ahead. None of these additions are conceptual advances, and are extremely straightforward additions to an existing method.
- The paper claims that "BRAID disentangles [input] dynamics from intrinsic dynamics", however the contribution of the inputs is not analyzed further at all - can one effectively disentangle input dynamics from intrinsic dynamics via this approach?
- The modeling strategy is multi-stage and quite involved, with multiple RNNs being trained without fully going into the utility of each one. The 'RNN_{fw}' models seem to be forecasting, but why is it necessary to have a separate RNN for forecasting when the 'RNN_1' model is a dynamical model that should be in theory capable of predicting m-steps forward in time?
- The R^2 should be reported throughout the paper instead of Pearson's; the R^2 is more standard in this field, and takes into account the predictability using the mean value of the signal.
- There is no attempt at interpretability of the underlying dynamics and the contribution from different sources as identified by this method.
- While the authors show that BRAID performs with higher behavior reconstructions than TNDM as shown in the Appendix, this is very much to be expected since TNDM does not optimize separately for behavior reconstruction, as BRAID does. Similarly, DPAD does not either (and does not take in inputs). However, these are very simple to add to both of these methods, and thus do not provide fair comparisons in their existing form.

**Questions:**

None noted / see above.

---

> ### Author Response · Authors · 2024-11-20
> **[W1]: BRAID's similarity to DPAD (Sani et al., 2024) and its conceptual advances**
>
> We thank the reviewer for their comment. Based on this comment and the reviewer’s comment 3 about *“why is it necessary to have a separate RNN for forecasting”*, it seems that the goal and conceptual advance of BRAID were not clearly communicated in our text. The primary goal of BRAID is to disentangle intrinsic dynamics from input dynamics, which is not even conceptually formulated in DPAD. Furthermore, we now show with new analyses expanded under W3 that doing so is not achieved by straightforward additions (e.g., simply adding input) to DPAD. Specifically, to do this, BRAID introduces two major conceptual advances in addition to multiple technical advances over DPAD.
>
> Conceptual advances:
>
> 1) We show that such disentanglement requires *two* RNNs learning fundamentally distinct dynamics and cannot be as accurately achieved with the single RNN used in DPAD: we need a separate generative RNN (RNN_fw) to learn the intrinsic dynamics in addition to another predictive RNN used in DPAD that integrates the neural observations for immediate state prediction. We now show that *even if* we add BRAID’s forecasting loss and/or inputs to the single RNN in DPAD, DPAD still learns the intrinsic dynamics inaccurately, as quantified by eigenvalues . See details and additional results below in response to [W3]. The distinction between predictor (RNN) and generative (RNN_fw) forms of dynamics is very fundamental and has parallels in linear dynamics systems theory, which we lay out in section 3.1 (see also the response to [W3]). DPAD only learns the former, while BRAID adds a distinct RNN to also learn the latter representations.
>
> 2) We show that incorporation of measured inputs necessitates a new computational step that uses a separate RNN to exclude behavior-specific dynamics before training, without which the learned behaviorally relevant dynamics may not even be neural, i.e., may not be present in neural activity (preprocessing stage, section 3.3). Indeed, if one trivially just adds input to DPAD as the reviewer notes for example by concatenating inputs to neural activity, DPAD will learn dynamics (i.e., find eigenvalues) that are *not* in neural activity and are just in behavior, because some dynamics in input just drive behavior and DPAD will have no way of knowing that. Note that this BRAID preprocessing step has no equivalent element in DPAD.
>
> In addition to the two major conceptual advances above, below are the technical advances over DPAD:
>
> Other technical advances:
>
> 3) The incorporation of a forecasting loss and showing that it is necessary to learn intrinsic dynamics. Note that having forecasting loss in the optimization is not at all supported in DPAD. As an additional baseline, we expand DPAD to support this and show that it is not enough to achieve the goals of BRAID.
>
> 4) Enabling the extraction of private dynamics in behavior (behavior-specific dynamics) by introducing a third set of RNNs (RNN3 and RNN3_fw) with the proper loss and proper links to the first two RNNs (see section 3.3). Again, this third set of RNNs has no equivalent in DPAD.
>
> In addition to the above, it is also worth noting that even adding the generative RNN_fw in two sections (stages 1 and 2, see section 3.2), with proper connections between the two sections of the predictor RNN was not straightforward and required new custom RNN cells with support for our orthogonal two RNN formulation (visualized in Fig. 1b) and was not possible as a simple extension of any existing method.
>
> Finally, at the end of the day, we use DPAD as a major baseline in both simulations and data and extensively show the benefits of BRAID over DPAD, with extended analyses as expanded on under W3 below. It is also worth noting that we did these extensive comparisons with DPAD even though according to [ICLR policy](https://iclr.cc/Conferences/2025/ReviewerGuide#:~:text=A%3A%20We%20consider%20papers%20contemporaneous,own%20work%20to%20that%20paper.), DPAD is considered concurrent work (because it was published at a peer reviewed venue on Sep. 6, 2024, which is 2 months after the July 1, 2024 cut-off for concurrency) and thus comparisons to it are not required by ICLR.

---

> ### Author Response · Authors · 2024-11-20
> **[W2]: Can one effectively disentangle input dynamics from intrinsic dynamics via this approach?**
>
> We appreciate this important comment by the reviewer. We show that BRAID can effectively disentangle input dynamics from intrinsic dynamics in our extensive simulations in which the ground-truth intrinsic dynamics, quantified by the eigenvalues, are known (Fig. 2c,g and Fig. A1d).  Our results show that BRAID more accurately recovers the intrinsic eigenvalues by explicitly modeling external inputs and learning the generative form of dynamics, compared to DPAD, which does not account for the external input. In real data, since ground-truth intrinsic eigenvalues are not known, we use the forecasting metrics that show the forward model has successfully learned and integrated both intrinsic and input-driven contributions as BRAID outperforms alternatives in forward prediction of both neural and behavioral data.

---

> ### Author Response · Authors · 2024-11-20
> **[W3]: Multiple RNNs' utilities and importance of a separate RNN for forecasting**
>
> This is a great question and lack of clarity on this may have also led to [W1] from the reviewer.
>
> We need multiple RNNs and a multi-stage learning algorithm because the goal here is to disentangle multiple subtypes of neural-behavioral dynamics. First, we have 3 sections/stages to learn and separate 3 types of neural-behavioral dynamics:
>
> 1) intrinsic behaviorally relevant neural dynamics (RNN1/RNN1_fw),
> 2) other intrinsic neural dynamics (RNN2/RNN2_fw),
> 3) behavior-specific dynamics (RNN3/RNN3_fw).
>
> Second, for each of the 3 groups, we learn both predictor (RNN) and generative (RNN_fw) representations of dynamics, because in general one representation cannot be deduced from the other one, so we need to learn both from data (see section 3.1).
>
> Our approach of having a separate RNN for each type of neural-behavioral dynamics and formulating the losses and stages of learning such that each RNN learns exactly the type and representation of dynamics that it is meant to is very beneficial. First, this way, each RNN is dedicated to a specific subtype of neural-behavioral dynamics, effectively disentangling the subtypes. Second, as we describe in methods section 3.2, the multi-stage learning of these 3 subtypes of dynamics allows us to *prioritize* the learning of intrinsic behaviorally relevant neural dynamics over other subtypes, such that we can learn them more accurately without them being confounded by other subtypes. Indeed, the primary focus here is the learning of this first subtype of dynamics. We have provided results showing utility of all these subtypes of dynamics (see Table 3, nx=16 and Fig. 3, for RNN1, Table 3, nx=64 for RNN2 and Fig. A2 for RNN3).
>
>
> The reviewer also points out a great question of why a separate RNN (RNN1_fw) is needed for forecasting. We apologize that this was not clear in our write up and will update the manuscript to clarify.
> While the predictor RNN is indeed a dynamical model, it learns a different representation of the dynamics compared with the generative RNN_fw:
>
> -  RNN performs a recurrent transformation that aggregates past input *and neural* observations to predict the latent state, whereas
> - RNN_fw generates future latent states only by aggregating inputs (and without relying on neural observations).
>
> These are two conceptually distinct processes and even their inputs are different so neither of them is able to replace the other even in terms of dimensionality of their inputs. Moreover, the recursive computation in these two cases are mathematically distinct (i.e., $A$ is not the same as $A_{fw}$).
>
> The distinction between predictor ($A$) and generative ($A_{fw}$) forms of dynamics is very fundamental and has parallels in linear system theory, where their exact relationship can be proven (section 3.1). Briefly, in a linear dynamical system model with state transition matrix $A_{fw}$, the generative form representation is how the states evolves *autonomously* with the recursion $A_{fw}$, while the predictor form representation shows how observations can be integrated to update state estimates via a Kalman filter (Van Overschee & De Moor, 1996). Similarly, in the nonlinear case, these two representations are not the same and furthermore their relationship is not known (unlike the linear case); therefore, it is necessary to directly learn generative models ($A_{fw}$) from the data, which BRAID does with its RNN_fw and forecasting objective. To make this more clear, in simulations, we now compare the recurrence learned by the predictor form RNN (instead of the generative form RNN_fw) with the true intrinsic dynamics and show that without learning a separate generative recursion, the predictor form RNN alone cannot learn the intrinsic dynamics (row 2 in table below). Moreover, we show that even if we extend DPAD to incorporate BRAID’s forecasting loss and inputs into DPAD (rows 3, 4 in table below), the single RNN in DPAD, which is in predictor form, still cannot find the intrinsic eigenvalues accurately. We will add the new results below for the ablation study inspired by the reviewer comment to the paper.
>
>
> Eigenvalue errors:
>
> Simulation with spiral manifold (Fig. 2a):
> | **Method**       | Log10 normalized eigenvalue error |
> |-------------------|------------------------------------------|
> | BRAID   |            **-1.3963±0.2551**              |
> | BRAID without RNN_fw            |       0.0635±0.2212               |
> |DPAD + m-step loss            |       0.0357±0.1587               |
> |DPAD + input + m-step loss            |       -0.6512±0.0354               |
>
>
>
> References:
> Peter Van Overschee and Bart De Moor. Subspace Identification for Linear Systems. Springer US, Boston, MA, 1996. ISBN 978-1-4613-8061-0. doi: 10.1007/978-1-4613-0465-4

---

> ### Author Response · Authors · 2024-11-20
> **[W4]: Reporting $R^2$ instead of Pearson's CC**
>
> We thank the reviewer for this suggestion. We will add R^2 values (at least for our main results) to the paper. We observed similar trends in terms of both R^2 and CC metrics, with BRAID’s R^2 values being *roughly* the square of the CC values, which suggests that no major scale or baseline mismatches existed for BRAID. In addition to R^2, we will also keep the CC results in the interest of being more easily comparable with those papers in the field that report CC. Please see the R^2 version of our main results in the temporary supplementary file (Table B.2).

---

> ### Author Response · Authors · 2024-11-20
> **[W5]: Interpretability of the underlying dynamics and the contribution from different sources**
>
> We thank the reviewer for this comment. As explained in response to [W2], as an attempt at interpreting intrinsic versus input dynamics, we have shown that BRAID successfully disentangles intrinsic and input dynamics in our simulations (eigenvalue errors e.g., in Fig. 2c.g). In interpreting the disengagement of dynamics, we focus on simulations because in simulations the ground truth for the sources of dynamics is known. When we show small errors in learning a specific type of dynamics, this is a direct interpretation of the learned dynamics by pointing at exactly what aspect (e.g., eigenvalues) in the simulated models were learned by BRAID. However, we agree with the reviewer that further analysis of the individual intrinsic vs input-driven components enabled by BRAID would be very interesting. We hope that our validation in simulations will encourage others to use our method to interpret their own datasets.

---

> ### Author Response · Authors · 2024-11-20
> **[W6]: Comparisons to TNDM and DPAD**
>
> We are not sure what the reviewer means by “optimize separately for behavior reconstruction”. First, if they mean the optional behavior-specific state ($x^{(3)}$ learned by RNN3) for behavior decoding, we should clarify that we never use these states in the decoding results as our aim is to decode behavior solely using neural activity and input (i.e., via the identified latent states in neural activity). Thus, the improvements in behavior decoding achieved by BRAID are not due to behavior-specific dynamics and the comparisons with TNDM and DPAD are fair. Second, if the reviewer means the use of behavior reconstruction during learning, both TNDM and DPAD, similar to BRAID, optimize for behavior reconstruction already using their behaviorally relevant latent states as one of their main objectives, with DPAD doing so in a separate optimization similarly to BRAID; so there is no need to add this to them. Third, to further ensure a fair comparison with TNDM, we also modified TNDM to accept sensory inputs alongside neural activity for inference of latent states. Even with this modification, BRAID outperformed the modified TNDM even though they both use the exact same signals for decoding (Table A3).

---

> > ### Comment · Reviewer_QzSK · 2024-11-26
> > **Thank you to the authors**
> >
> > Thank you for your extensive responses. While I appreciate the additional analyses and comparisons, and will increase my score to reflect these very helpful additions, I maintain that the authors have a very complex learning structure with multiple stages and multiple networks. They may perform better at certain tasks than their baselines, but the complexity hinders the interpretability of the resulting networks. In fact, the authors do not discuss the interpretability in much detail at all - what do we gain by applying this on the experimental dataset apart from ability to decode more time-steps ahead? How would this be different from just training a model to decode the behavior 8 time steps ahead directly from the neural data - what is the additional utility or interpretability that one gets out of inferring an underlying latent state? The notation of RNN1, RNN2, etc., is also very difficult to follow. In short, while the necessity of having one model for modeling the effect of the input on the state and another to model intrinsic dynamics is now clarified, the overall utility of having in essence 6 different models and their corresponding interpretability is somewhat opaque.

---

> > > ### Author Response · Authors · 2024-11-27
> > >
> > > We thank the reviewer for their response. We are encouraged that the reviewer finds our new analyses very helpful and will increase their score. We also thank them for their remaining questions about complexity and interpretability of our approach, which are important points that we now clarify below.
> > >
> > > The primary goal of our work is not to simply decode behavior but rather to disentangle various subtypes of intrinsic neural-behavioral dynamics (i.e., latent states), which can indeed facilitate interpretability in neuroscience investigations. These subtypes are:
> > >
> > > i) intrinsic behaviorally relevant neural dynamics
> > >
> > > ii) neural-specific intrinsic dynamics
> > >
> > > iii) behavior-specific dynamics
> > >
> > > So our goal is to not only learn latent representations of neural-behavioral data, but also make sure that these fundamentally different subtypes of dynamics are disentangled in the latent representations. The disentangling of these 3 subtypes is where our 3 stages of learning come into play:
> > >
> > > Stage 1: exclusively learns subtype i.
> > >
> > > Stage 2: exclusively learns subtype ii, by modeling the residual neural activity that is not learned by stage 1.
> > >
> > > Stage 3: exclusively learns subtype iii, by modeling the residual behavior data that is not learned by stage 1. Moreover, a preprocessing step ensures that this subtype (iii) is not learned by stage 1.
> > >
> > > The disentanglement of these 3 subtypes was why we designed 3 stages because, otherwise,  a single-stage mixed-cost optimization (e.g., as in baselines U-BRAID and mmPLRNN)  can mix up these subtypes in the same latent dimensions, hence not achieving disentanglement and hindering interpretability. Finally, in each of the above 3 stages, the model consists of 2 RNNs (predictor and generator) to disentangle intrinsic dynamics of the subtype in that stage from input dynamics that are driving it, as the reviewer also notes. The generator RNN learns the intrinsic dynamics by forecasting, starting from the predictor RNN’s embedding at each time.
> > >
> > > Regarding our notation, we agree that it can be difficult to follow. To help address this, we have now added the **new table A.1** (copied below for convenience), to summarize the notation in one place:
> > >
> > > | **Stage**              | **Models**         | **Functionality**                                         |
> > > |---------------------|--------------------|----------------------------------------------|
> > > | 1 | $RNN1, RNN1_{fw}$ | Intrinsic behaviorally relevant neural dynamics         |
> > > | 2 | $RNN2, RNN2_{fw}$  | Residual neural-specific intrinsic dynamics             |
> > > | 3 | $RNN3, RNN3_{fw}$  | Residual behavior-specific dynamics not encoded in neural activity |
> > >
> > > Regarding complexity of this approach, we would also like to note that stages 2 and 3 are complementary to stage 1 and their use is *optional* depending on the application. For example, one may choose to just run stage 1 if they only care about extracting embeddings for the first subtype of dynamics, in which case they will only use stage 1 (which will learn only $RNN1$ and $RNN1_{fw}$). As another example, in our real data analyses, we do not use stage 3 (we do not learn $RNN3$ and $RNN3_{fw}$) because our main goal is to learn intrinsic dynamics encoded in neural activity, not just to improve behavior decoding performance.
> > >
> > > The above explains the **utility** of each stage of learning and each part of the model and **how** each serves our goal of disentangling different subtypes of intrinsic neural-behavioral dynamics. We now address the **interpretability** and  **why** questions in a separate comment next.

---

> > > ### Author Response · Authors · 2024-12-03
> > >
> > > As today is the last day of the discussion period, we wanted to follow up with the reviewer to make sure that they have seen our response regarding their follow-up interpretability and complexity questions. We would appreciate it if the reviewer could kindly take a look at this recent response. We again thank the reviewer for their comments and are encouraged that the reviewer finds our new analyses very helpful and mentions that they will increase their score.

---

> ### Author Response · Authors · 2024-11-27
>
> The reviewer asks:
> >How would this be different from just training a model to decode the behavior 8 time steps ahead directly from the neural data
>
> Our new ablation studies in **new table A.3** show that simply training a model (e.g., DPAD) to decode behavior multiple steps ahead does not learn the correct intrinsic neural dynamics, which is a primary goal here. In this analysis, we added multi-step ahead loss to DPAD, and showed in simulations where the ground truth is known that it still does not learn the intrinsic dynamics. BRAID in contrast accurately learns the intrinsic dynamics using its distinct predictor-generator architecture (figure 1b).
>
>
> Next, we clarify **why** the disentanglement of subtypes of dynamics is of interest and what **interpretability** it provides. The reviewer asks:
> >what is the additional utility or interpretability that one gets out of inferring an underlying latent state?
>
> First, one of the key goals of learning disentangled latent representations, or disentangled representation learning (Wang et al 2024), is to make the learned embedding interpretable. The fact that we know which latent state dimensions are associated with each subtype of dynamics makes the model more interpretable. One of the ways this interpretability can be useful is now shown in **new Fig. A4**. In this figure, we have visualized the trajectories (temporal evolution) of the latent states that BRAID learns for the first subtype of dynamics in our experimental data. These latent state trajectories are more congruent with reach directions compared with those extracted from other methods, suggesting that BRAID has been more successful in learning and disentangling the intrinsic behaviorally relevant neural dynamics of motor cortex (M1). Learning more interpretable and behavior related latent representation has been a key goal for many impactful methods in neuroscience (Churchland et al 2012, Kobak et al 2016, Sani et al 2021, etc).
>
> Second, the latent states in BRAID model the *temporal evolution* of a neural population as a dynamical system, for example the eigenvalues associated with each state specify the decay and oscillations in the temporal evolution. In the “computation through neural population dynamics” view (Vyas et al, 2020), the neural population performs computations through these dynamics to generate behavior, e.g., movement. As such, the study of these latent states in dynamical system models (e.g., BRAID) can be used to study neural computations (Remington et al., 2018; Shenoy et al., 2013). The contribution of our work to this framework is to help address the major problem of disentangling the dynamics that are intrinsic to the population from those that are due to external sensory inputs. Doing so is important, for example, to identify cortical dynamics that are more autonomous/intrinsic (Seely et al. 2016; Shenoy et al., 2021;  Sauerbrei et al., 2020; Vyas et al 2020; Vahidi et al 2024). Indeed, our work allows not only for such intrinsic vs. input disentanglement, but also for dissociating the 3 subtypes of intrinsic neural-behavioral dynamics as stated in our response above. As demonstrated in our simulations (Figs. 2, A1), our method enables accurate learning of the intrinsic dynamics as quantified by the eigenvalues. Also, as explained in our response above, the new ablation analysis (**new table A.3**) shows that the BRAID architecture is critical in learning such accurate intrinsic dynamics.
>
>
> The above descriptions, references and results clarify **why** in BRAID we aim to dissociate the three subtypes of intrinsic dynamics in neural-behavioral data, and to further disentangle input dynamics from them. These also explain the ***interpretability*** offered by doing so. Please let us know if you have any other questions or would like further clarification on any point. We again thank the reviewer for all their important comments and are encouraged that the reviewer finds our new analyses *"very helpful"* and *"will increase"* their score.
>
>
> Please see the next comment for the references.

---

> ### Author Response · Authors · 2024-11-27
>
> **References**:
>
> Mark M Churchland, John P Cunningham, Matthew T Kaufman, Justin D Foster, Paul Nuyujukian, Stephen I Ryu, and Krishna V Shenoy. Neural population dynamics during reaching. Nature, 487 (7405):51–56, 2012.
>
> Dmitry Kobak, Wieland Brendel, Christos Constantinidis, Claudia E Feierstein, Adam Kepecs, Zachary F Mainen, Xue-Lian Qi, Ranulfo Romo, Naoshige Uchida, and Christian K Machens. Demixed principal component analysis of neural population data. elife, 5:e10989, 2016.
>
> Evan D Remington, Seth W Egger, Devika Narain, Jing Wang, and Mehrdad Jazayeri. A dynamical systems perspective on flexible motor timing. Trends in cognitive sciences, 22(10):938–952, 2018.
> Omid G Sani, Hamidreza Abbaspourazad, Yan T Wong, Bijan Pesaran, and Maryam M Shanechi. Modeling behaviorally relevant neural dynamics enabled by preferential subspace identification. Nature Neuroscience, 24(1):140–149, 2021.
>
> Britton A Sauerbrei, Jian-Zhong Guo, Jeremy D Cohen, Matteo Mischiati, Wendy Guo, Mayank Kabra, Nakul Verma, Brett Mensh, Kristin Branson, and Adam W Hantman. Cortical pattern generation during dexterous movement is input-driven. Nature, 577(7790):386–391, 2020.
>
> Jeffrey S Seely, Matthew T Kaufman, Stephen I Ryu, Krishna V Shenoy, John P Cunningham, and Mark M Churchland. Tensor analysis reveals distinct population structure that parallels the different computational roles of areas m1 and v1. PLoS computational biology, 12(11):e1005164, 2016.
>
> Krishna V Shenoy and Jonathan C Kao. Measurement, manipulation and modeling of brain-wide neural population dynamics. Nature communications, 12(1):633, 2021.
> Krishna V Shenoy, Maneesh Sahani, and Mark M Churchland. Cortical control of arm movements: a dynamical systems perspective. Annual review of neuroscience, 36(1):337–359, 2013.
>
> Parsa Vahidi, Omid G Sani, and Maryam M Shanechi. Modeling and dissociation of intrinsic and input-driven neural population dynamics underlying behavior. Proceedings of the National Academy of Sciences, 121(7):e2212887121, 2024.
>
> Saurabh Vyas, Matthew D Golub, David Sussillo, and Krishna V Shenoy. Computation through neural population dynamics. Annual review of neuroscience, 43(1):249–275, 2020.
>
> Xin Wang, Hong Chen, Si’ao Tang, Zihao Wu, and Wenwu Zhu. Disentangled Representation Learning. IEEE Transactions on Pattern Analysis and Machine Intelligence, 46(12):9677–9696, 2024.

---

### Author Response · Authors · 2024-11-20
**Global Response**

We thank the reviewers for taking the time to review our submission and for providing constructive comments, suggestions, and discussions regarding our work. We are pleased that the reviewers found that our work is “interesting” and “novel” and addresses “a very relevant question” in the field, and appreciated the soundness of our results, our “comprehensive” experiments and ablations, and the quality of presentation.


We have addressed all comments and questions from the reviewers by performing several new analyses and have provided the results either inline in our individual responses to reviewers (e.g., as tables and text), and/or as part of a new temporary supplementary PDF file. We will integrate these new results into the manuscript and upload a revised version by the end of the discussion period. In the meantime, please refer to the new PDF file uploaded in the Supplementary Materials to see the new results. Should there be any remaining questions or concerns, we would be happy to clarify them, perform additional analyses to address them, and to engage in further discussion.

---

> ### Author Response · Authors · 2024-11-25
>
> We have now updated the manuscript by incorporating all the new analyses and discussions from the temporary supplementary file into it. The uploaded updated manuscript now addresses all comments from all reviewers. We once again thank all reviewers for the valuable and constructive feedback, which we believe has significantly improved our manuscript.

---

### Meta-Review · Area_Chair_stji · 2024-12-20

**Metareview:**

This paper presents a novel approach to latent dynamical modeling of neural data. The authors claim that previous techniques treat neural circuits as autonomous dynamical systems, when in fact, they are strongly shaped by external inputs. Here, the authors present a technique for learning a dynamical model that separates out the contribution from external inputs and intrinsic recurrent dynamics in the circuit. Their technique, Behaviorally Relevant Analysis of Intrinsic Dynamics (BRAID), uses a series of recurrent neural networks: (1) a shared recurrent neural network that outputs both neural activity and behavior, (2) a network but for neural dynamics only, and (3) a network for behavioral dynamics only. The authors support their claim that, unlike previous models, BRAID can disentangle the inputs from the intrinsic dynamics by showing better correlation with the neural and behavioral activity in validation datasets as compared to baseline models.

The strengths of this paper are that it is well-motivated, the writing is clear, and the experiments are reasonably convincing in supporting the claims. The weaknesses are that the model is fairly complicated and the novelty a bit limited. However, on balance, the authors did a good job of responding to the concerns of the reviewers and the model likely has utility for the neuroscience community. As such, a decision of accept (poster) was reached.

**Additional Comments On Reviewer Discussion:**

The reviews were mostly constructive and engaged. The authors worked hard to address the reviewer concerns, and three out of the four reviewers engaged with them and raised their scores in response. One reviewer did not engage, and left their score well below the acceptance threshold. However, due to the lack of engagement by this reviewer, the AC discounted their opinion to some degree when crafting their meta-review.

---

### Decision · Program_Chairs · 2025-01-22

Accept (Poster)